# MEDCALC-BENCH: Evaluating Large Language Models for Medical Calculations

**Nikhil Khandekar**[*1, 4], **Qiao Jin**[*1], **Guangzhi Xiong**[*2], **Soren Dunn**[3, 4], **Serina S Applebaum**[5],
**Zain Anwar**[7], **Maame Sarfo-Gyamfi**[8], **Conrad W Safranek**[5], **Abid A Anwar**[6], **Andrew Zhang**[9],
**Aidan Gilson**[5], **Maxwell B Singer**[5], **Amisha Dave**[5], **Andrew Taylor**[5],
**Aidong Zhang**[2], **Qingyu Chen**[5], and **Zhiyong Lu**[†1]

[1]National Library of Medicine, National Institutes of Health, [2]University of Virginia, [3]University of
Illinois at Urbana Champaign, [4]Lapis Labs, [5]Yale University School of Medicine, [6]University of
Illinois College of Medicine at Chicago, [7]Rosalind Franklin University Chicago Medical School,
[8]Howard University College of Medicine, [9]University of Chicago Pritzker School of Medicine

## Abstract

Current benchmarks for evaluating large language models (LLMs) in medicine
are primarily focused on question-answering involving domain knowledge and
descriptive reasoning. While such qualitative capabilities are vital to medical
diagnosis, in real-world scenarios, doctors frequently use clinical calculators that
follow quantitative equations and rule-based reasoning paradigms for evidence-
based decision support. To this end, we propose MEDCALC-BENCH, a first-of-
its-kind dataset focused on evaluating the medical calculation capability of LLMs.
MEDCALC-BENCH contains an evaluation set of over 1000 manually reviewed
instances from 55 different medical calculation tasks. Each instance in MEDCALC-
BENCH consists of a patient note, a question requesting to compute a specific
medical value, a ground truth answer, and a step-by-step explanation showing how
the answer is obtained. While our evaluation results show the potential of LLMs in
this area, none of them are effective enough for clinical settings. Common issues
include extracting the incorrect entities, not using the correct equation or rules for a
calculation task, or incorrectly performing the arithmetic for the computation. We
hope our study highlights the quantitative knowledge and reasoning gaps in LLMs
within medical settings, encouraging future improvements of LLMs for various
clinical calculation tasks. [1]

## 1 Introduction

Large language models (LLMs) such as GPT [2, 31], Gemini/PaLM [1, 47], and Llama [50, 51]
have been successfully applied to a variety of biomedical tasks [29, 38, 48, 49], including, but not
limited to question answering [27, 30, 42], clinical trial matching [23, 57, 58, 62], and medical
document summarization [40, 46, 52]. However, most of these tasks have a limited evaluation of
domain knowledge and qualitative reasoning ability of LLMs, as demonstrated by the commonly
used medical benchmarks such as MedQA [21], PubMedQA [22], and MedMCQA [33]. While

---

*Equal contribution. †Correspondence: `zhiyong.lu@nih.gov`.

[1]MEDCALC-BENCH is publicly available at: `https://github.com/ncbi-nlp/MedCalc-Bench`.

38th Conference on Neural Information Processing Systems (NeurIPS 2024) Track on Datasets and Benchmarks.

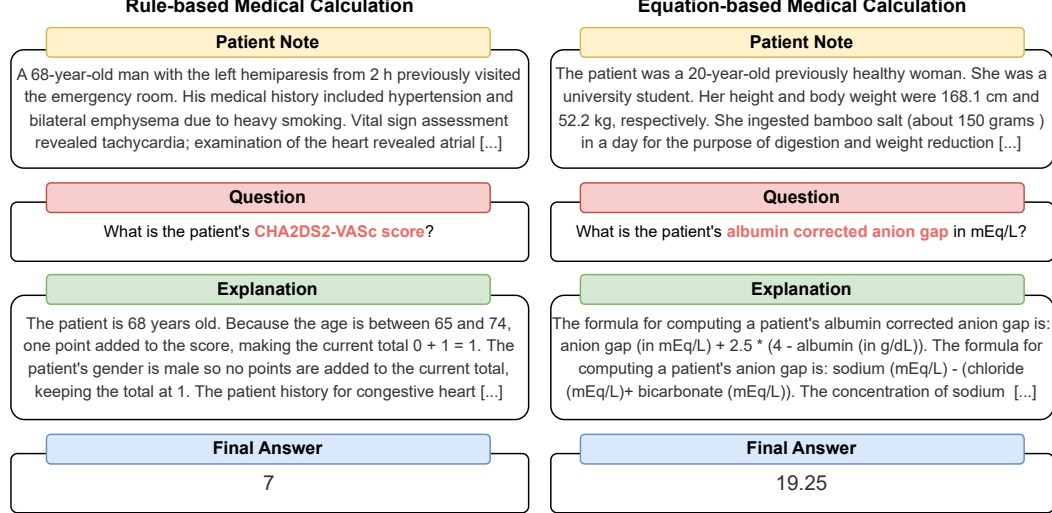

Figure 1: Example instances of the MEDCALC-BENCH dataset.

quantitative tools such as medical calculators are frequently used in clinical settings [8, 14], currently there is no benchmark evaluating the medical calculation capabilities of LLMs.

Medical calculators are statistical tools derived from high-quality clinical studies, serving various purposes, including metric conversions [7], disease diagnosis [18], and prognosis prediction [11]. Figure 1 shows two examples of medical calculators. To accurately compute requested medical scores, the model needs to have three non-trivial capabilities: (1) possessing the knowledge of the rules or equations for the medical calculation task, (2) identifying and extracting the values of the relevant parameters within a long patient note, and (3) conducting the arithmetic computation for the task correctly.

In this work, we propose MEDCALC-BENCH, a first-of-its-kind dataset for evaluating the medical calculation capabilities of LLMs. To construct MEDCALC-BENCH, we first curated 55 common medical calculation tasks from MDCalc[2]. Then, we compiled Open-Patients, a collection of over 180k publicly available patient notes, and identified the notes that can be used for each calculation task. Finally, we collected over 1k instances for MEDCALC-BENCH, where each instance contains: (1) a patient note, (2) a question requesting to compute a specific medical value, (3) a ground truth answer, and (4) a step-by-step explanation of the computation process.

Using MEDCALC-BENCH, we conducted systematic evaluations of various LLMs, including the state-of-the-art proprietary models such as GPT-4 [31], open-source LLMs such as Llama [51] and Mixtral [20], as well as biomedical domain-specific PMC-LLaMA [59] and MEDITRON [4]. Our experimental results show that most of the tested models struggle in the task. GPT-4 achieved the best baseline performance of only 50.9% accuracy using one-shot chain-of-thought prompting. By analyzing the types of errors made by LLMs, we found that the models suffer mostly from insufficient medical calculator knowledge in the zero-shot setting. To mitigate this issue, we add a one-shot exemplar in the prompt, showing the model how to apply the requested medical equations or rules. Our analysis revealed additional issues in extracting calculator-related attributes and in arithmetic computations. These results can provide insights into future improvement in the medical calculation capabilities of LLMs.

In summary, the contributions of our study are threefold:

- We manually curated MEDCALC-BENCH, a novel dataset of over 1k instances for evaluating the capabilities of LLMs across 55 different medical calculation tasks.

---

[2]https://www.mdcalc.com/#Popular

- We conducted comprehensive evaluations on MEDCALC-BENCH with various open and closed-source LLMs. Our results show that all current LLMs are not yet ready for medical calculations, with the best accuracy of only 50.9% achieved by GPT-4.

- Our error analysis reveals the insufficiency of calculator knowledge in LLMs, as well as their deficiencies in attribute extraction and arithmetic computation for medical calculation.

## 2 MEDCALC-BENCH

### 2.1 Calculation Task Curation

MEDCALC-BENCH covers 55 different calculators. These were all listed as "popular" on MDCalc, the most commonly used online medical calculator website by clinicians [9]. As shown in Figure 1, they fall into two major categories, **rule-based** calculation (19 calculators) and **equation-based** calculation (36 calculators).

Rule-based calculators typically contain a list of criteria, where each criterion is a condition of a specific medical attribute. An instance of this would be the HEART score calculator [43], which takes in both numerical attributes such as the patient's age (e.g., if the patient is older than 65 years, add two points; if the patient's age is between 45 and 64, add one point; and zero points otherwise) and categorical variables such as the presence of significant ST elevation (adding two points if present; zero points otherwise). The final answer for these calculators will be a discrete value after taking the sum of the sub-scores.

Like rule-based calculators, equation-based calculators also take in both categorical (e.g., gender, race) and numerical variables (e.g., creatinine concentration, age, and height). However, equation-based calculators follow a specific formula to output a decimal, date, or time given the attributes instead of additively combining sub-scores for each criterion. An instance of an equation-based calculator would be the MDRD GRF equation [26]. This equation computes the patient's eGFR, which is a function of the patient's gender and race as coefficients in addition to the patient's creatinine concentration. The only equation-based calculators which do not output a decimal are Estimated Due Date (EDD), Estimated Date of Conception (EDC), and Estimated Gestational Age (EGA). These three calculators compute a date (for EDC, EGA) or a time (for EGA) instead.

For each instance, MEDCALC-BENCH also provides a natural language explanation for how the final answer is computed. We implement template-based explanation generators for each of the 55 calculators. These templates first list the numerical and categorical variable values, and then plug them in to show how the final answers are obtained. The implementation details can be found in supplementary materials.

### 2.2 Dataset Instance Collection

In this section, we describe how patient notes and answers were collected for the 55 different calculation tasks in MEDCALC-BENCH. We aimed to collect at most 20 notes for each calculator. Specifically, the patient notes were collected using the following three-step pipeline.

**(1) Note collection and attribute extraction.** We compiled Open-Patients[3], a collection of over 180k public patient notes, including anonymized real case reports from PMC-Patients [60], case vignettes in MedQA-USMLE [21], synthetic cases in TREC Clinical Decision Support Tracks [41, 36] and TREC Clinical Trials Tracks [37]. Using GPT-3.5-Turbo, we identified patient notes for each calculator based on its eligibility criteria. We then used GPT-4 to extract the attribute values needed for each calculator from the eligible notes.

**(2) Data verification and enrichment.** For each of the patient notes for a given calculator, the extracted values from GPT-4 were verified and corrected by one individual with a medical background. After the verification, 34 of the 55 calculators had at least one eligible note with the extracted attributes

---

[3]Publicly available at `https://huggingface.co/datasets/ncbi/Open-Patients`.

needed for computation. Hence, the remaining 21 calculators without any eligible notes were either synthesized with a template or were handwritten by an individual with a medical background.

Of these 21 calculators, 10 are equation-based calculators, for which we generated 20 synthesized notes using a template. The other 11 calculators are rule-based and we employed the same individuals with medical background to synthesize the patient notes and record the ground-truth values for the needed attributes.

**(3) Answer and explanation generation.** After obtaining patient notes with the extracted values, for each of the 55 calculators, we generated step-by-step explanations to derive the final answers. Specifically, we implemented templates for each calculator to generate the natural language explanations. From these three steps, we curated 1047 instances for MEDCALC-BENCH, each of which contains a patient note, a question, along with a ground-truth explanation and final answer.

## 2.3 Dataset Characteristics

Table 1 shows the statistics of MEDCALC-BENCH and the different calculator sub-types. The equation-based calculators have between 1 to 7 attributes, while the rule-based calculators have between 3 to 31 attributes. Thus, it may require a varying number of reasoning steps to solve different tasks in our dataset.

Table 1: Statistics of MEDCALC-BENCH. Inst.: instance; Avg.: average; Attr.: attribute; Q.: question; L: length.

| | #Tasks | #Inst. | Avg. L of Note | Avg. L of Q. | Min Attr. | Max Attr. | Avg. Attr. | Example Calculation |
|---|---|---|---|---|---|---|---|---|
| **Equation-based Calculation Tasks** | | | | | | | | |
| Lab | 19 | 327 | 891.0 | 22.3 | 2 | 7 | 3.6 | LDL Concentration |
| Physical | 12 | 240 | 419.3 | 20.8 | 1 | 3 | 2.0 | QTc (Bazett Formula) |
| Date | 3 | 60 | 25.3 | 67.0 | 2 | 2 | 2.0 | Estimated Due Date |
| Dosage | 2 | 40 | 31.4 | 31.0 | 2 | 6 | 4.0 | Morphine Equivalents |
| **Rule-based Calculation Tasks** | | | | | | | | |
| Risk | 12 | 240 | 422.1 | 14.9 | 5 | 31 | 11.5 | Caprini Score for VTE |
| Severity | 4 | 80 | 262.6 | 11.0 | 3 | 20 | 7.7 | Pneumonia Severity Idx |
| Diagnosis | 3 | 60 | 625.6 | 15.0 | 3 | 9 | 5.3 | PERC Rule for PE |
| **Overall** | 55 | 1047 | 529.7 | 21.9 | 1 | 31 | 5.4 | – |

Our dataset evaluates three distinct capabilities required for medical calculation:

**(1) Recall of medical calculation knowledge.** The first required capability is to successfully recall the formulas from seven different domains shown in Table 1. As mentioned above, medical calculators can have various sub-types with varying number of attributes. Hence, LLMs are challenged to know every detail about medical equations or rules to solve a question in MEDCALC-BENCH.

**(2) Extraction of relevant patient attributes.** The second required capability is the extraction of correct attributes from patient notes, given the noises in the long context of over 500 words on average. LLMs are required to extract both numerical and categorical attributes. The medical context complicates such extractions, with the existence of multiple synonyms (e.g., both HbA1c and glycohemoglobin denote the same entity) and the requirement of determining the presence of certain medical cases without explicitly being stated (e.g., a blood pressure of 160/100 mmHg indicates the presence of hypertension). Hence, LLMs require both medical knowledge and clinical reasoning to solve questions in this dataset.

**(3) Arithmetic computation of the final results.** The third required capability is the computation of final results, especially the derivation of scores through multi-step reasoning. While datasets like

GSM-8k [6] have tested the arithmetic calculation capability of LLMs, MEDCALC-BENCH presents a more challenging task. Our dataset requires LLMs to fully understand the sequence and dependencies among multiple medical equations or rules, some of which need to be chained together, to obtain the correct answer. Additionally, MEDCALC-BENCH also contains some exponential computations that are not covered by other math datasets.

Overall, we believe that MEDCALC-BENCH serves as a comprehensive benchmark that goes beyond testing the internal medical calculation knowledge of LLMs. This dataset also tests general-purpose skills such as attribute extraction and arithmetic computation in a more challenging domain-specific setting.

# 3 Evaluation

## 3.1 Settings

To establish the baseline performance in MEDCALC-BENCH, we experiment with eight different LLMs under three common prompting strategies. We categorize the eight LLMs into three groups: **Medical domain-specific** LLMs include PMC-LLaMA-13B [59] and MEDITRON-70B [4]; **Proprietary** LLMs include GPT-4 [31] and GPT-3.5 [32]; **Open-source** LLMs, including 8B and 70B Llama 3 [51], as well as Mistral-7B [19] and Mixtral-8x7B [20].

Similarly, we consider three prompting strategies: **Zero-shot Direct Prompting**: In this setting, the LLM is prompted to directly output the answer without any explanation; **Zero-shot Chain-of-Thought (CoT) Prompting**: In this setting, the LLM is prompted to first generate step-by-step rationale and then generate the answer [56]; **One-shot CoT Prompting**: In this setting, the LLM is provided with an exemplar of the corresponding calculation task. The exemplar is manually curated and contains the patient note, question, and the output consisting of the step-by-step explanation and final answer value.

Based on the output type, we have three different evaluation settings: (1) For all rule-based calculators, the final answer must be the exact same as the ground-truth answer, (2) For equation-based calculators that are lab tests, physical tests, and dosage conversion calculators, the predicted answer must be within 5% of the ground-truth answer, (3) For date-based equation calculations, the predicted dates should exactly match the ground truth.

## 3.2 Main Results

Table 2 presents our evaluation results of various LLMs on the 1047 instances from MEDCALC-BENCH. From the table, we can observe the diverse performance of the models in different settings. In general, LLMs tend to perform better with the help of CoT prompting and one-shot learning, as evidenced by the improved accuracy for each LLM shown in the table. Among all LLMs compared, GPT-4 achieves the best performance in all three settings. In the zero-shot direct promoting setting, GPT-4 has a mean accuracy of 20.82% on the task. By leveraging its own reasoning ability, the performance of GPT-4 can be improved to 37.92%. Incorporating external medical knowledge from a one-shot demonstration further increases its accuracy to 50.91%. Similar patterns can also be observed in many other LLMs, such as LLama 3 and Mixtral.

In addition to the general trend across different settings, the table also shows how various types of LLMs perform differently on our MEDCALC-BENCH test. While GPT-4 performs the best in our evaluation, the open-source Llama 3-70B model shows a competitive performance that is close to GPT-4. In both zero-shot direct prompting and zero-shot CoT prompting settings, Llama 3-70B achieves mean accuracies that are comparable to the results of GPT-4. However, GPT-4 significantly outperforms LLama 3-70B with the one-shot demonstration, which reflects its superior in-context learning capability for medical calculation. Moreover, by comparing Llama 3-8B/Mistral-7B with Llama 3-70B/Mixtral-8x7B, we find larger LLMs generally perform better on the medical calculation tasks, corresponding to the empirical scaling laws [17, 25]. Interestingly, the 70B MEDITRON cannot

Table 2: MEDCALC-BENCH accuracy of different systems. All numbers are in percentages. Phys.: Physical; Sev.: Severity; Diag.: Diagnosis; Avg.: Average; ± std for all results are shown. Each std. is computed with the following formula: $\sqrt{(\text{accuracy} * (1 - \text{accuracy})/(n)}$, where $n$ is the total number of patient notes for a given subcategory for rule or equation-based calculators.

| Model | Size | Equation | | | | Rule-based | | | Avg. |
|---|---|---|---|---|---|---|---|---|---|
| | | Lab | Phys. | Date | Dosage | Risk | Sev. | Diag. | |
| **Zero-shot Direct Prompting** | | | | | | | | | |
| PMC-LLaMA [59] | 13B | 0.0 ±0.00 | 0.0 ±0.00 | 0.0 ±0.00 | 0.0 ±0.00 | 0.0 ±0.00 | 0.0 ±0.00 | 0.0 ±0.00 | 0.0 ±0.00 |
| MEDITRON [4] | 70B | 3.7 ±0.01 | 8.3 ±0.02 | 5.0 ±0.03 | 0.0 ±0.00 | 7.5 ±0.02 | 5.0 ±0.02 | 13.3 ±0.04 | 6.2 ±0.01 |
| Mistral [19] | 7B | 10.7 ±0.02 | 18.3 ±0.02 | 3.3 ±0.02 | 0.0 ±0.00 | 4.6 ±0.01 | 3.8 ±0.02 | 13.3 ±0.04 | 9.8 ±0.01 |
| Mixtral [20] | 8x7B | 12.2 ±0.02 | 23.3 ±0.03 | 5.0 ±0.03 | 7.5 ±0.04 | 12.9 ±0.02 | 7.5 ±0.03 | 16.7 ±0.05 | 14.2 ±0.01 |
| Llama 3 [51] | 8B | 10.7 ±0.02 | 19.2 ±0.03 | 3.3 ±0.02 | 5.0 ±0.03 | 12.5 ±0.02 | 8.8 ±0.03 | 25.0 ±0.06 | 13.1 ±0.01 |
| Llama 3 [51] | 70B | 18.0 ±0.02 | 33.3 ±0.03 | 8.3 ±0.04 | 12.5 ±0.05 | 15.8 ±0.02 | 13.8 ±0.04 | 33.3 ±0.06 | 20.8 ±0.01 |
| GPT-3.5 [32] | N/A | 17.1 ±0.02 | 35.0 ±0.03 | 13.3 ±0.04 | 5.0 ±0.03 | 12.9 ±0.02 | 6.3 ±0.03 | 18.3 ±0.05 | 18.8 ±0.01 |
| GPT-4 [31] | N/A | 14.4 ±0.02 | 34.6 ±0.03 | 38.3 ±0.06 | 15.0 ±0.06 | 14.6 ±0.02 | 15.0 ±0.04 | 20.0 ±0.05 | 20.8 ±0.01 |
| **Zero-shot CoT Prompting** | | | | | | | | | |
| PMC-LLaMA [59] | 13B | 0.0 ±0.00 | 0.0 ±0.00 | 0.0 ±0.00 | 0.0 ±0.00 | 0.0 ±0.00 | 0.0 ±0.00 | 0.0 ±0.00 | 0.0 ±0.00 |
| MEDITRON [4] | 70B | 0.0 ±0.00 | 0.0 ±0.00 | 3.3 ±0.02 | 0.0 ±0.00 | 0.0 ±0.00 | 0.0 ±0.00 | 3.3 ±0.02 | 0.4 ±0.00 |
| Mistral [19] | 7B | 10.1 ±0.02 | 14.6 ±0.02 | 1.7 ±0.02 | 0.0 ±0.00 | 9.6 ±0.02 | 7.5 ±0.03 | 25.0 ±0.06 | 10.8 ±0.01 |
| Mixtral [20] | 8x7B | 22.6 ±0.02 | 40.0 ±0.03 | 6.7 ±0.03 | 17.5 ±0.06 | 11.3 ±0.02 | 21.3 ±0.05 | 15.0 ±0.05 | 22.4 ±0.01 |
| Llama 3 [51] | 8B | 16.5 ±0.02 | 25.0 ±0.03 | 1.7 ±0.02 | 7.5 ±0.04 | 11.3 ±0.02 | 13.8 ±0.04 | 26.7 ±0.06 | 16.4 ±0.01 |
| Llama 3 [51] | 70B | 33.9 ±0.03 | 66.3 ±0.03 | 25.0 ±0.06 | 20.0 ±0.06 | 18.3 ±0.02 | 16.3 ±0.04 | 36.7 ±0.06 | 35.5 ±0.01 |
| GPT-3.5 [32] | N/A | 20.5 ±0.02 | 45.0 ±0.03 | 11.7 ±0.04 | 17.5 ±0.06 | 13.3 ±0.02 | 10.0 ±0.03 | 31.7 ±0.06 | 23.7 ±0.01 |
| GPT-4 [31] | N/A | 26.3 ±0.02 | 71.3 ±0.03 | 48.3 ±0.06 | 40.0 ±0.08 | 27.5 ±0.03 | 15.0 ±0.04 | 28.3 ±0.06 | 37.9 ±0.01 |
| **One-shot CoT Prompting** | | | | | | | | | |
| PMC-LLaMA [59] | 13B | 5.2 ±0.01 | 10.4 ±0.02 | 8.3 ±0.04 | 2.5 ±0.02 | 7.1 ±0.02 | 1.3 ±0.01 | 11.7 ±0.04 | 7.0 ±0.01 |
| MEDITRON [4] | 70B | 22.9 ±0.02 | 39.6 ±0.03 | 31.7 ±0.06 | 15.0 ±0.06 | 20.4 ±0.03 | 15.0 ±0.04 | 31.7 ±0.06 | 26.3 ±0.01 |
| Mistral [19] | 7B | 11.0 ±0.02 | 30.4 ±0.03 | 6.7 ±0.03 | 0.0 ±0.00 | 16.3 ±0.02 | 6.3 ±0.03 | 18.3 ±0.05 | 16.1 ±0.01 |
| Mixtral [20] | 8x7B | 28.1 ±0.02 | 50.8 ±0.03 | 8.3 ±0.04 | 22.5 ±0.07 | 21.3 ±0.03 | 8.8 ±0.03 | 33.3 ±0.06 | 29.2 ±0.01 |
| Llama 3 [51] | 8B | 34.9 ±0.03 | 35.4 ±0.03 | 3.3 ±0.02 | 2.5 ±0.02 | 20.0 ±0.03 | 11.3 ±0.04 | 41.7 ±0.06 | 27.1 ±0.01 |
| Llama 3 [51] | 70B | 41.6 ±0.03 | 56.3 ±0.03 | 30.0 ±0.06 | 22.5 ±0.07 | 27.5 ±0.03 | 27.5 ±0.05 | 45.0 ±0.06 | 39.5 ±0.02 |
| GPT-3.5 [32] | N/A | 30.9 ±0.03 | 59.2 ±0.03 | 41.7 ±0.06 | 15.0 ±0.06 | 23.3 ±0.03 | 17.5 ±0.04 | 35.0 ±0.06 | 34.9 ±0.01 |
| GPT-4 [31] | N/A | 51.7 ±0.03 | 77.5 ±0.03 | 46.7 ±0.06 | 37.5 ±0.08 | 33.8 ±0.03 | 27.5 ±0.05 | 53.3 ±0.06 | 50.9 ±0.02 |

beat Mistral-7B in the zero-shot settings, which can be explained by its poor instruction-following capability as the officially released model has not been instruction-tuned. With the additional demonstration in a one-shot setting, MEDITRON effectively learns the task and shows an improved performance close to Mixtral-8x7B.

It can also be observed from the table that the results for different subtasks in MEDCALC-BENCH present distinct patterns. For example, the performance of GPT-4 on the physical value calculation task is improved by 36.67% by adding CoT prompting, while including an additional demonstration only further increases its accuracy by 6.25%. In contrast, GPT-4 performance is improved by 8.33% and 25%, respectively, on the diagnosis calculation task, with the help of CoT prompting and one-shot demonstration. This result reflects that GPT-4 already contains certain medical knowledge concerning physical value calculation, thus the CoT prompting alone can significantly enhance its performance on such tasks. Nevertheless, the diagnosis calculation information is insufficient in its parametric knowledge, so the extra one-shot demonstration offers better help compared to the CoT prompting. Such an analysis enables us to have insights into the capabilities and limitations of LLMs on various medical calculation tasks, suggesting their different use cases in real-world applications.

## 4 Discussion

In this section, we provide an in-depth analysis of errors made by LLMs on MEDCALC-BENCH.

### 4.1 What types of errors can LLMs make in MEDCALC-BENCH?

We categorize four types of errors that LLMs can make in our dataset: **Type A (knowledge errors):** the model does not have the correct knowledge of the equation or rules used in the medical calculation task; **Type B (extraction errors):** the model extracts the wrong parameters from the patient note; **Type C (computation errors):** the model conducts the arithmetic incorrectly; **Type D (other errors):** all other cases of errors. Specific examples of the first three error types are shown in Table 3.

Table 3: Examples of three error types in MEDCALC-BENCH. The incorrect rationale, # comments, and evidence are highlighted.

| Type | Question | Patient Note | Error of GPT-4 Zero-shot CoT |
|---|---|---|---|
| **A - Knowledge Error** | What is the patient's Wells' criteria score for Deep Vein Thrombosis? | A 77-year-old man was diagnosed with RA in 2006 with positive anti-CCP antibodies and RF; MTX was used but was changed to GLM in 2016 (the recent dose interval was 6–7 weeks; the last dose was 5 weeks before vaccine administration) and maintained [...] | The Wells' criteria for Deep Vein Thrombosis (DVT) includes [...] the patient has a brain hemorrhage, which is more likely than a DVT. The patient scores 1 point for paralysis and 1 point for an alternative diagnosis being less likely. # Wrong point assignment [...] |
| **B - Extraction Error** | What are the number of criteria met for the PERC Rule for Pulmonary Embolism (PE)? | A 61-year-old male presented to our emergency department complaining of [...] A computerized tomography with contrast of the chest revealed multiple filling defects within the pulmonary arterial system, indicative of bilateral acute on chronic thromboembolic disease. [...] | The PERC Rule for Pulmonary Embolism (PE) includes eight criteria: [...] The patient is older than 50 years, his pulse rate is less than 100 bpm, his SaO2 is less than 94%, there is no mention of hemoptysis, estrogen use, prior DVT or PE # The patient had PE, recent surgery or trauma, or unilateral leg swelling. [...] |
| **C - Computation Error** | What is the patient's LDL cholestrol concentration? Please output your answer in terms of mg/dL. | A 37-year-old male presented with progressive swelling of feet and facial puffiness for 6 months. On physical examination, he had mild pallor, moderate pedal edema and cloudy cornea with a peripheral arcus in both eyes [...] serum haptoglobin of 64 mg/dl (normal range: 70–200 mg/dl), total cholesterol 215 mg/dl, [...] | The patient's [...] LDL = Total cholesterol - HDL - (Triglycerides / 5). The values needed for this calculation are given in the note: Total cholesterol is 215 mg/dL, HDL is 10 mg/dL, and Triglycerides are 385 mg/dL. Answer: LDL = 215 - 10 - (385 / 5) = 142 # Wrong computation mg/dL |

It should be noted that these errors are not independent of each other. For example, LLMs usually recall the relevant calculator knowledge first, and then extract the relevant parameters, and finally conduct the computation. If the model recalls the calculator incorrectly, it is highly likely that it cannot extract the correct set of relevant parameters. Hence, we only consider the earliest error if there are multiple error types (e.g., if the model has error types A and B, then the error type will be A).

## 4.2 What errors do different LLMs make?

To analyze the errors made by different LLMs, we utilize GPT-4 to classify their error types by comparing the LLM output to the ground truth in MEDCALC-BENCH. We manually evaluate the annotations of 200 randomly sampled explanation errors, and find the accuracy of GPT-4 error classifier to be 89%. As such, we apply it to analyze the mistakes of all CoT prompting results.

Table 4 shows the distribution of error types in different settings. Under the zero-shot CoT setting, most of the errors (more than 50%) belong to Type A in all LLMs, suggesting that recalling the correct equations or rules for the corresponding medical calculation task is the biggest challenge when no exemplar is provided. While Type A error is dominant under the zero-shot setting, its error rate varies in different LLMs, e.g. 0.96 in PMC-LLaMA and 0.35 in GPT-4. Such a difference reflects the diverse levels of medical calculation knowledge acquired by various LLMs.

Unlike the distributions in the zero-shot setting, errors that occurred with the one-shot CoT prompting have less than 50% being categorized as Type A, which is consistently observed in different LLMs. This shows the effectiveness of the one-shot exemplar in providing the background rule or equation needed for medical calculation. With the decrease in Type A errors, more Type B and Type C errors are captured in the wrong answers. This reveals the deficiencies of current LLMs in attribute extraction and arithmetic computation, which are required capabilities to perform real-world medical calculations.

Table 4: Error type distribution of LLMs on MEDCALC-BENCH. Numbers in parentheses denote the relative proportions. Arrows indicate the proportion changes from zero-shot to one-shot learning.

| Model | Type A Error | Type B Error | Type C Error | Type D Error | Error Rate |
|---|---|---|---|---|---|
| *Zero-shot CoT Prompting* | | | | | |
| PMC-LLaMA-13B | 0.96 (96%) | 0.03 (3%) | 0.00 (0%) | 0.01 (1%) | 1.00 |
| MEDITRON-70B | 0.97 (97%) | 0.00 (0%) | 0.02 (2%) | 0.00 (0%) | 1.00 |
| Mistral-7B | 0.72 (80%) | 0.11 (12%) | 0.06 (7%) | 0.00 (0%) | 0.89 |
| Mixtral-8x7B | 0.55 (71%) | 0.11 (14%) | 0.09 (11%) | 0.02 (3%) | 0.78 |
| Llama 3-8B | 0.60 (72%) | 0.11 (13%) | 0.13 (15%) | 0.00 (0%) | 0.84 |
| Llama 3-70B | 0.43 (67%) | 0.10 (16%) | 0.11 (17%) | 0.00 (0%) | 0.64 |
| GPT-3.5 | 0.38 (50%) | 0.24 (31%) | 0.13 (17%) | 0.02 (2%) | 0.76 |
| GPT-4 | 0.35 (57%) | 0.19 (30%) | 0.08 (13%) | 0.00 (0%) | 0.62 |
| *One-shot CoT Prompting* | | | | | |
| PMC-LLaMA-13B | 0.42 (46%↓) | 0.31 (34%↑) | 0.17 (19%↑) | 0.01 (1%–) | 0.91 |
| MEDITRON-70B | 0.24 (33%↓) | 0.23 (32%↑) | 0.26 (35%↑) | 0.01 (1%↑) | 0.74 |
| Mistral-7B | 0.32 (38%↓) | 0.33 (40%↑) | 0.17 (20%↑) | 0.01 (1%↑) | 0.83 |
| Mixtral-8x7B | 0.27 (38%↓) | 0.23 (33%↑) | 0.19 (27%↑) | 0.01 (2%↓) | 0.71 |
| Llama 3-8B | 0.25 (34%↓) | 0.17 (24%↑) | 0.29 (40%↑) | 0.02 (2%↑) | 0.73 |
| Llama 3-70B | 0.20 (34%↓) | 0.12 (20%↑) | 0.23 (39%↑) | 0.05 (8%↑) | 0.60 |
| GPT-3.5 | 0.23 (36%↓) | 0.20 (30%↓) | 0.20 (31%↑) | 0.02 (2%–) | 0.65 |
| GPT-4 | 0.20 (40%↓) | 0.13 (27%↓) | 0.16 (33%↑) | 0.00 (0%–) | 0.49 |

## 4.3 Limitations and Future Work

While our study provides a first-of-its-kind dataset to evaluate the medical calculation capabilities of various LLMs, there are several main limitations that can be improved in future work: (1) Due to the difficulty of manual verification of each instance in MEDCALC-BENCH, our dataset is limited in size, containing only 1047 instances in total. (2) We had a limited number of annotators with a medical

background and so only one individual could verify the GPT-4 parameter extractions. While there is no subjectivity for extracting numerical attribute values, there can be disagreement on descriptive attribute values such as determining whether a patient has renal disease for the HAS-BLED calculator [28]. Hence, there may be a bias based on the individual's training and so there may be some subjectivity in these values. (3) While we saw a significant improvement in model performance with the one-shot exemplar, benchmarking the model with few-shot instances may have further increased the accuracy. However, curating such patient notes for rule-based calculators would have been difficult, given the labor-intensiveness of having to synthesize patient notes often requiring many attributes.

For future work, we will have follow-up updates on the dataset to improve its quality, adding more instances for each calculator. There is also room for benchmarking with advanced methods that have shown improvement on GSM-8k [6] such as step-by-step PPO [54] and scaling test-time inference [44]. We leave such explorations as future work for researchers to enhance the medical computational capabilities of LLMs.

# 5 Related Work

Table 5: Comparison of different datasets for LLM evaluation. Medical: tasks for medical evaluation; Knowledge: dataset tests knowledge to a particular domain; Qualitative (Qual) Reasoning: dataset tests qualitative reasoning; Comput.: dataset requires computation (i.e., quantitative reasoning); Non-MCQ: questions which have a single answer and without the use of multiple choices. Explanation: dataset provides a step-by-step reasoning.

| | Medical | Knowledge | Qual. Reasoning | Comput. | Non-MCQ | Explanation |
|---|---|---|---|---|---|---|
| MedQA [21] | ✔ | ✔ | ✔ | ✗ | ✗ | ✗ |
| MedMCQA [33] | ✔ | ✔ | ✔ | ✗ | ✗ | ✗ |
| PubMedQA [22] | ✔ | ✗ | ✔ | ✗ | ✗ | ✗ |
| MMLU [15] | ✔ | ✔ | ✔ | ✗ | ✗ | ✗ |
| GSM8k [6] | ✗ | ✗ | ✗ | ✔ | ✔ | ✔ |
| MATH [16] | ✗ | ✗ | ✗ | ✔ | ✔ | ✔ |
| OpenMedCalc [13] | ✔ | ✔ | ✔ | ✔ | ✔ | ✗ |
| AgentMD [24] | ✔ | ✔ | ✔ | ✔ | ✔ | ✗ |
| CalcQA [61] | ✔ | ✔ | ✔ | ✔ | ✔ | ✗ |
| Sci-Bench [55] | ✗ | ✔ | ✗ | ✔ | ✔ | ✔ |
| Tutor-Eval [5] | ✗ | ✔ | ✔ | ✔ | ✔ | ✔ |
| **MEDCALC-BENCH** | ✔ | ✔ | ✔ | ✔ | ✔ | ✔ |

## 5.1 Language Model Evaluations in Medicine

Existing datasets for evaluating LLMs in biomedicine [10] have primarily focused on verbal reasoning through multiple choice questions such as PubMedQA [22], MedQA [21], MedMCQA [33], and the medical questions in MMLU [15]. However, these datasets are mainly focused on qualitative reasoning instead of quantitative computation. Additionally, the format of multi-choice questions does not reflect the actual clinical settings where a single answer or response must be determined without any options provided. In this work, we introduce MEDCALC-BENCH, the first dataset that measures the quantitative reasoning capabilities of LLMs in medicine in a realistic setting where the LLM must determine the answer by itself without the support of answer choices.

## 5.2 Language Model Evaluations in Computation

Many efforts have been made to evaluate the mathematical and computation capability of LLMs in various settings. GSM8k [6] and MATH [16] are two examples which focus on pure mathematical problems. However, such datasets with general settings may not reflect LLM performance in domain-specific applications. In contrast, Sci-Bench[55] and Tutor-Eval [5] both include step-by-step explanations for domain-specific computation, but their focus is on college-level science as opposed to the medical field.

Additionally, one of the key features of language agents is the capability to use tools [34, 45, 53, 63], such as code interpreters [3, 12] and external APIs [35, 39]. As such, these capabilities have been applied to medical computation tasks. Although OpenMedCalc [13], CalcQA [61], and AgentMD [24] use medical calculators to augment LLMs, their evaluations are based on small-scale or automatically constructed datasets. MEDCALC-BENCH is much larger than their evaluation datasets and contains both natural language explanations as well as final numeric answers.

Hence, MEDCALC-BENCH serves as the first dataset for medical-focused calculations with explanations. A full comparison of various datasets can be found in Table 5.

## 6  Conclusion

In conclusion, this study introduces MEDCALC-BENCH, the first dataset designed to evaluate the capabilities of LLMs for medical calculations. Our evaluations show that while LLMs like GPT-4 exhibit potential, none are reliable enough for clinical use. The error analysis highlights areas for improvement, such as knowledge recall and computational accuracy. We hope our work serves as a call to further improve LLMs and make them more suitable for medical calculations.

## Acknowledgments and Disclosure of Funding

This research was supported by the NIH Intramural Research Program, National Library of Medicine. Additionally, the contributions made by Soren Dunn were done using the Delta advanced computing and data resource which is supported by the National Science Foundation (award OAC tel:2005572) and the State of Illinois. Delta is a joint effort of the University of Illinois Urbana-Champaign (UIUC) and its National Center for Supercomputing Applications (NCSA).

## Ethics Statement

For curating the patient notes in MEDCALC-BENCH, we only use publicly available patient notes from published case report articles in PubMed Central and clinician-generated anonymous patient vignettes. As such, **no identifiable personal health information is revealed in this study**.

While MEDCALC-BENCH is designed to evaluate the medical calculation capabilities of LLMs, it should be noted that the dataset is not intended for direct diagnostic use or medical decision-making without review and oversight by a clinical professional. Individuals should not change their health behavior solely on the basis of our study.

## Broader Impacts

As described in Sec 1, medical calculators are commonly used in the clinical setting. With the rapidly growing interest in using LLMs for domain-specific applications, healthcare practitioners might directly prompt chatbots like ChatGPT to perform medical calculation tasks. However, the capabilities of LLMs in these tasks are currently unknown. Since healthcare is a high-stakes domain and wrong medical calculations can lead to severe consequences, including misdiagnosis, inappropriate treatment plans, and potential harm to patients, it is crucial to thoroughly evaluate the performance of LLMs in medical calculations. Surprisingly, the evaluation results on our MEDCALC-BENCH dataset show that all the studied LLMs struggle in the medical calculation tasks. The most capable model GPT-4 achieves only 50% accuracy with one-shot learning and chain-of-thought prompting. As such, our study indicates that **current LLMs are not yet ready to be used for medical calculations**.

It should be noted that while high scores on MEDCALC-BENCH do not guarantee excellence in medical calculation tasks, failing in this dataset indicates that the models must not be considered for such purposes at all. In other words, we believe that passing MEDCALC-BENCH should be a necessary (but not sufficient) condition for a model to be used for medical calculation.

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
