# Supplementary Materials for MEDCALC-BENCH

## Contents

# A  Dataset Curation - Additional Details

## A.1  Extraction Process

As mentioned in Section 2, the patient notes used for MEDCALC-BENCH come from three main sources: (1) Publicly available patient notes from Open-Patients, (2) Notes synthesized from templates (Section A.2), (3) Notes synthesized from templates.

The majority of instances of MEDCALC-BENCH comes from Open-Patients, a dataset of 180k publicly available patient notes which are aggregated from four different sources as shown in the table below:

Table 1: Publicly available patient notes which make up Open-Patients

| Source | # Patients | Average Token Length | Note Type |
|---|---|---|---|
| TREC Clinical Decision Support | 90 | 105.1 | This track consists of datasets of 30 patient notes each for three separate years from 2014-2016. The motivation of this track was to challenge participants to obtain relevant articles that can help answer potential questions for a particular patient note. The patient notes 2014 and 2015 are synthetic patient notes hand-written by individuals with medical training, but the 2016 dataset consists of real patient summaries coming from electronic health records. |
| TREC Clinical Trials | 125 | 137.7 | This track consists of 125 patient notes, where 50 notes are from the year of 2021 and 75 notes are from the year of 2022. This track was meant to have participants retrieve previous clinical trials from ClinicalTrials.gov that best match the symptoms described in the patient note. The notes from both tracks are synthetic notes written by individuals with medical training meant to simulate an admission statement from an electronic health record (EHR). |
| MedQA-USMLE | 12,893 | 135.8 | Questions from a multiple-choice form a professional medical board exam which can include patient summaries and will ask questions about particular issues based on the summary. |
| PMC-Patients | 167,034 | 484 | Patient summaries extracted from case reports from PubMedCentral |

This collection is released on HuggingFace under the CC-BY-SA 4.0 license: `https://huggingface.co/datasets/ncbi/Open-Patients/tree/main`.

Using the Open-Patients dataset, we curated notes for each calculator using this three-stage pipeline:

**1.** We used GPT-3.5-turbo to identify eligible patient notes for each calculator. We did this by shortlisting the notes that had at least one relevant required for a given calculator.

**2.** With the shortlisted notes for each calculator, we then used GPT-3.5-turbo to check whether the note met the eligibility criteria for using the calculator based on the requirements provided by MDCalc.

**3.** Lastly, using the clinically eligible notes for each calculator from the second step, we used GPT-4 to extract the remaining attributes. We kept the notes that had all of the numeric parameters needed for each calculator. These notes were also filtered to have enough categorical variables inferred such that 50% of the total number of attributes were present for a given patient note.

The attribute extractions of these notes were then verified for by authors of this paper.

**Extraction 1 Prompt:**

For more details on step 1, we provided a set of 32 parameters which cover at least one attribute needed for each of the 55 calculators. For each note in Open-Patients, we applied the prompt shown below to determine which of the 32 parameters could be extracted from each note. From this, we received a dictionary of the 32 parameter IDs and their extracted values for each note.

You are a helpful assistant for extracting the values of medical parameters from a patient note. Here is the list of parameters to consider:

| Parameter ID | Parameter Name | Type | Values & Units |
|---|---|---|---|
| 3 | weight | numerical | kg, lbs |
| 4 | height | numerical | m, cm, in, ft |
| 5 | creatinine | numerical | mg/dL, μmol/L |
| 13 | Systolic Blood Pressure | numerical | mm Hg |
| 16 | Albumin | numerical | g/dL, g/L |
| 20 | Heart Rate or Pulse | numerical | beats per minute |
| 28 | Total cholesterol | numerical | mg/dL, mmol/L |
| 112 | Cerebrovascular disease history | categorical | True, False |
| 133 | Absence of cough or coryza | categorical | True, False |
| 60 | Initial troponin | categorical | less than or equal to normal limit, between the normal limit or up to three times the normal limit, greater than three times normal limit |
| 61 | Aspartate aminotransferase | numerical | U/L |
| 66 | Temperature | numerical | degrees fahrenheit, degrees celsius |
| 73 | Sodium | numerical | mEq/L, mmol/L |
| 91 | Glucose | numerical | mg/dL, mmol/L |
| 93 | Blood Urea Nitrogen (BUN) | numerical | mg/dL, mmol/L |
| 102 | respiratory rate | numerical | breaths per minute |
| 106 | White blood cell count | numerical | count/μL, count/L, count/mm³ |
| 6 | History of Congestive Heart Failure | categorical | True, False |
| 31 | Bilirubin | numerical | mg/dL |
| 63 | Platelet count | numerical | count (per billion)/L, count (per thousand)/μL |
| 114 | Partial pressure of oxygen (PaO$_2$) | numerical | mm Hg |
| 108 | FiO$_2$ | numerical | % |
| 44 | Previously documented Deep Vein Thrombosis | categorical | True, False |
| 8 | Stroke | categorical | True, False |
| 9 | Transient Ischemic Attacks History | categorical | True, False |
| 7 | Hypertension history | categorical | True, False |
| 105 | Hematocrit | numerical | % |
| 107 | Glasgow Coma Score | numerical | nan |
| 23 | Hemoptysis | categorical | True, False |
| 32 | international normalized ratio | numerical | ratio |
| 215 | Ideal Body Weight (IBW) | numerical | kg |
| 47 | History of ischemic heart disease | categorical | True, False |

Please check the parameters one-by-one, and output a JSON dict in the following format: Dict{Str(Parameter_ID): Str(value)}. This JSON should report the parameter id and a value for the parameter in the patient note (if available) for all 32 parameters. If there are multiple values for a given measurement or attribute, then please use the value recorded based on when the patient note was written. You should not be using values that the patient had post-treatment or values from a patient's history in the past. For parameters whose 'Type' is 'categorical', the exact name provided in the 'Parameter Name' column might not be mentioned inside the patient note. For such categorical variables, you should do your best to infer the value taken by the parameter. Additionally, for parameters whose "Type" is categorical, please select from one of values separated by commas in the Values & Units column. Examples: {"8": "True"}, {"164": "Uncomplicated"}. If there is no mention of a categorical parameter by the exact name and it cannot easily be inferred, then the value of the parameter should be "Not Mentioned" as the value, e.g. {"26": "Not Mentioned"} or {"90": "Not Mentioned"}.

If the 'Type' of the parameter is 'numerical' and the parameter is mentioned, please extract the exact values and units from the patient note and separate them by spaces. Examples: {'4': 5 'ft' 4 'in'}, {'3': 41.2 kg}, {'13': 67 mm Hg}, {'20': 117 beats per minute}. For numerical parameters whose label is a concentration (mass/volume), the mass units MUST be one of the following: [mol, mmol, µmol, pmol, g, mg, µg, kg, mEq] and the units of volume MUST be one of the following: ['L', 'dL', 'mL', 'µL', 'mm$^3$', 'cm$^3$', 'm$^3$']. Examples: {'16': 15.5 g/L}, {'5': 2.34 mg/dL}. The only exception to this rule is for platelet count and white blood cell count whose mass unit is 'count' and the volume unit can be one of ['L', 'dL', 'mL', 'µL', 'mm$^3$', 'cm$^3$', 'm$^3$']. Examples: {'106': 15034 count/µL}, {'63': 1750000 count/L]}. For all other numeric parameters whose labels are not in the form of concentrations (mass/volume), you MUST use one of the EXACT label names provided in the Values & Type column. Examples: {'66': 39 degrees celsius}. Numerical attributes which are not mentioned in the patient note should have their value set to 'Not Mentioned.' Please output a JSON dict in the mentioned format, specifying the value for each parameter ID in the provided list.

Your task is to extract the 32 medical attributes and their values provided to you from the following patient note:

A 23-year-old previously healthy male presented to the ED with complaints of a headache that was gradual in onset and had been present for the prior 24 hours. He noted some lightheadedness and dizziness while standing, which prompted him to present to the ED for evaluation. He was febrile to 100.5 degrees Fahrenheit (F) and tachycardic to 110 beats per minute (bpm). The remainder of his physical exam was grossly unremarkable with no meningeal signs or focal neurologic deficits. He was provided antipyretics and intravenous (IV) fluids with complete resolution of his symptoms and discharged home with a diagnosis of viral syndrome. Two days later, he returned to the ED with complaints of continued headache and fever. He recalled a dry, tickling throat which was brief and self-limited in the prior two days. He was tachycardic, but afebrile on exam. With the exception of his tachycardia, his physical exam was again unremarkable without an identifiable infectious source. Laboratory evaluation demonstrated a bandemia of 8% (reference range 0) as well as mild transaminitis with alanine aminotransferase (ALT) 177 units per liter (U/L) (reference range 17) and aspartate aminotransferase (AST) 171 U/L (reference range 12Ž01339). His rapid heterophile antibody test was positive. He was discharged home with precautions to avoid contact sports and to have repeated liver function tests performed by his primary care provider. Three days after his second ED visit, he returned with jaundice, dark urine, and with continued fever and fatigue. He denied sore throat, cough, chest pain, abdominal pain, vomiting, diarrhea, hematuria, dysuria, or rash. He was again febrile with a temperature of 100.9Ŏ0b0 F and a pulse rate of 109 bpm. There was noticeable scleral icterus and diffuse jaundice. He was also noted to have multiple, palpable, posterior cervical lymph nodes.Laboratory evaluation was notable for a leukocytosis of 14.8 Ŏ0d710$^3$ cells per microliter (mcL) (reference 4.0Ž01310.5) with lymphocytic predominance of 24% and thrombocytopenia of 99Ŏ0d710$^3$ cells/mcL (reference range 150Ž013450). Comprehensive metabolic panel was notable for mild hyponatremia of 133 millimoles (mmol) per L (reference range 136Ž013145 mmol/L), total bilirubin of 7.93 milligrams per deciliter (mg/dL) (reference 0.15Ž013100), direct bilirubin of 6.9 mg/dL (reference range <0.2Ž01310.3), alkaline phosphatase of 198 U/L (reference range 40Ž013129), ALT of 753 U/L (reference range 17Ž01363), and AST 692 U/L (reference range 12Ž01339). Coagulation studies were within normal limits. Acetaminophen level was negative at <1.5 micrograms per milliliter (reference range 10Ž01330). Hepatitis serologies were notable for a reactive hepatitis B virus core antibody, non-reactive hepatitis B core antibody IgM, positive hepatitis B surface antibody, and negative hepatitis B surface antigen consistent with immunity due to natural infection. Hepatitis C antibody was non-reactive. Human immunodeficiency virus testing was negative. Blood cultures were also negative. EBV heterophile antibodies were positive.A formal right upper quadrant ultrasound demonstrated a mildly enlarged liver with normal contour. The gallbladder was visualized and noted to be contracted. The gallbladder wall was noted to be mildly thickened with a measurement of 0.34 centimeters. There was no evidence of cholelithiasis. The patient was admitted to the hospital for supportive care and further laboratory evaluation. He was provided IV fluids, and liver function tests were trended every six hours. Liver enzymes gradually decreased and his jaundice resolved. His thrombocytopenia was thought to be related to acute hepatitis. Coagulation studies remained within normal limits. He was discharged from the hospital with a diagnosis of cholestatic hepatitis secondary to EBV. He followed up with internal medicine and had serial liver function tests over the subsequent weeks."

Remember to use the appropriate units for numerical parameters and to use one of the values mentioned in the Values & Units column for categorical parameters. There should only be one extracted value for each parameter and never a list. For numerical parameters which are not explicitly mentioned, please set the value as 'Not Mentioned'. For categorical parameters whose value cannot be inferred, please set the value as 'Not Mentioned.' You should output all of the parameter IDs and their values inside a JSON dict.

```
Output

{
    "3": "Not Mentioned",
    "4": "Not Mentioned",
    "5": "Not Mentioned",
    "13": "110 mm Hg",
    "16": "Not Mentioned",
    "20": "110 beats per minute",
    "28": "Not Mentioned",
    "112": "Not Mentioned",
    "133": "Not Mentioned",
    "60": "Not Mentioned",
    "61": "171 U/L",
    "66": "100.5 degrees Fahrenheit",
    "73": "133 mmol/L",
    "91": "Not Mentioned",
    "93": "Not Mentioned",
    "102": "Not Mentioned",
    "106": "14.8 x10^3 cells/mcL",
    "6": "Not Mentioned",
    "31": "0.34 centimeters",
    "63": "99 x10^3 cells/mcL",
    "114": "Not Mentioned",
    "108": "Not Mentioned",
    "44": "Not Mentioned",
    "8": "Not Mentioned",
    "9": "Not Mentioned",
    "7": "Not Mentioned",
    "105": "Not Mentioned",
    "107": "Not Mentioned",
    "23": "Not Mentioned",
    "32": "Not Mentioned",
    "215": "Not Mentioned",
    "47": "Not Mentioned"
}
```

**Step 2: Determine Clinical Eligibility**

From step 1, we curated notes for each calculator that had at least one attribute needed for the calculator. We then determined whether the patient was clinically eligible based on the criteria provided by MDCalc.

Using GPT-3.5-turbo, we applied the following prompt to find the eligible notes for each calculator based on the shortlisted set of notes for each calculator in step 1. Shown below is an example for checking the eligibility for the CURB-65 calculator:

## User Prompt

**User Prompt:**
Patient Note: A 70-year-old Caucasian male initially presented to the emergency department (ED) of our hospital with fever and chills, which began one hour prior to his presentation. He also reported nausea and a productive cough with greenish sputum. The patient had been previously admitted to our hospital for pneumonia a month prior to his presentation and was discharged to a rehabilitation facility for three weeks. His past medical history was significant for end stage renal disease (ESRD) with dialysis dependence, failed kidney transplant, coronary artery disease (CAD) status with four drug-eluting stents (DES), type 2 diabetes mellitus, chronic pneumonia, and hypertension. On admission, his vital signs were as follow: temperature 39.4°C, blood pressure 87/55 mm Hg, pulse 100 beats per minute (bpm), oxygen saturation 88% on room air and respiratory rate 18 breaths per minute. Physical examination revealed coarse breath sounds bilaterally on auscultation, a left upper arm arteriovenous fistula as well as multiple superficial abrasions bilaterally on his lower extremities. His laboratory findings were as follows: white blood cells 15.12 K/µL, platelets 104 K/µL, sodium 134 mmol/L, blood urea nitrogen 39 mg/dl, creatinine 4.5 mg/dl, lactic acid 2.77 mmol/L and procalcitonin 25.24 ng/ml. Urine analysis was not performed because the patient was anuric. His chest X-ray showed evidence of small effusions and bibasilar airspace disease which were indicative of atelectasis or infection. An electrocardiogram showed normal sinus rhythm, a rate of 99 bpm, normal axis and no significant ST abnormalities, with paroxysmal ventricular contractions (PVCs). Blood cultures were drawn, and the patient received ceftriaxone and piperacillin/tazobactam empiric antibiotics and intravenous (IV) fluids. Gram stain of the blood specimen showed gram-negative rods. He was initially admitted to the progressive care unit (PCU) with cardiac telemetry. He was found to have second-degree atrioventricular (AV) block and was subsequently transferred to the intensive care unit (ICU) for closer management. On the second day of admission, blood cultures grew Pasteurella multocida. The organism was sensitive to beta-lactams including ampicillin, amoxicillin, amoxicillin/clavulanic acid and ampicillin/sulbactam. The patient was treated with IV cefepime due to his comorbidities. At this time, the patient was further questioned about recent animal contact. He stated that he lived with his wife and 14 cats and admitted the cats usually scratched him when they laid on him. His subsequent hospital stay was unremarkable. He was discharged home six days after admission with amoxicillin/clavulanic acid for two weeks. The patient returned to the outpatient clinic two weeks after discharge and had made an uneventful recovery.

Does this patient meet the eligibility criteria for CURB-65 Score? Simply check based on whether the patient's condition meets the criteria and DO NOT base your answer based on whether the note mentions the required attributes needed for computing a score with that calculator. You can have an explanation if you would like, but at the end, your response MUST include either ELIGIBILITY: Yes or ELIGIBILITY: No somewhere inside your note.

ELIGIBILITY: Yes

Based on the information provided in the patient note, the patient meets the eligibility criteria for using the CURB-65 Score calculator. The calculator is used to risk stratify a patient's community-acquired pneumonia in the emergency department setting. Although some specific values required for computation of the CURB-65 Score are missing in the patient note, such as confusion, blood urea nitrogen level, respiratory rate, and age, the information provided in the note indicates that the patient has pneumonia symptoms with a productive cough, fever, and chills. Additionally, the patient has risk factors such as advanced age, comorbid conditions including end-stage renal disease, diabetes, and coronary artery disease, as well as abnormal vital signs including low blood pressure, tachycardia, and low oxygen saturation.

It is important to note that the CURB-65 Score is used to assess the severity of pneumonia and determine the need for hospitalization, but additional clinical judgment and evaluation are necessary for appropriate management and treatment decisions.

**Prompt for Step 3:**

Lastly, after curating the eligible notes for each calculator, we then extracted the remaining parameters needed for the calculator. We provide the following example of how the parameter extraction from a patient note is done for the CURB-65 Score calculator:

---

**System Prompt**

You are a helpful assistant for extracting the values of medical parameters needed for CURB-65 Score for Pneumonia Severity from a patient note. Here is a table of parameters to consider:

| Parameter ID | Parameter Name | Type | Values & Units | Description |
|---|---|---|---|---|
| 102 | respiratory rate | numerical | breaths per minute | nan |
| 13 | Systolic Blood Pressure | numerical | mm Hg | nan |
| 14 | Diastolic Blood Pressure | numerical | mm Hg | nan |
| 176 | Confusion | categorical | True, False | nan |
| 2 | age | numerical | years, months, weeks, days | nan |
| 93 | Blood Urea Nitrogen (BUN) | numerical | mg/dL, mmol/L | nan |

Please check the parameters one-by-one, and output a JSON dictionary in the following format: Dict{Str(Parameter_ID): Str(value)}. This JSON should report the parameter id and a value for the parameter in the patient note (if available) for all the parameters mentioned. If there are multiple values for a given measurement or attribute, then please use the value recorded based on when the patient note was written. You should not be using values that the patient had post-treatment or values from a patient's history in the past.

For parameters whose 'Type' is 'categorical', the exact name provided in the 'Parameter Name' column might not be mentioned inside the patient note. For such categorical variables, you should do your best to infer the value taken by the parameter. If available, you should use the information in the 'Description' column to help you get more context about a variable and it's values which may help with inferring the value. Additionally, for parameters whose "Type" is categorical, please select from one of values separated by commas in the Values & Units column. Examples: {"8": "True"}, {"164": "Uncomplicated"}. If there is no mention of a categorical parameter by the exact name and it cannot easily be inferred, then the value of the parameter should be "Not Mentioned" as the value, e.g. {"26": "Not Mentioned"} or {"90": "Not Mentioned"}. Numerical parameters should never be inferred if explicitly not given and should be taken to be "Not Mentioned," if the parameter is not mentioned in the patient note.

If the 'Type' of the parameter is 'numerical' and the parameter is mentioned, please extract the exact values and units from the patient note and separate them by spaces. Examples: {'4': 5 'ft' 4 'in'}, {'3': 41.2 kg}, {'13': 67 mm Hg}, {'20': 117 beats per minute}. For numerical parameters whose label is a concentration (mass/volume), the mass units MUST be one of the following: [mol, mmol, μmol, pmol, g, mg, μg, kg, mEq] and the units of volume MUST be one of the following: ['L', 'dL', 'mL', 'μL', 'mm$^3$', 'cm$^3$', 'm$^3$']. Examples: {'16': 15.5 g/L}, {'5': 2.34 mg/dL}. The only exception to this rule is for platelet count and white blood cell count whose mass unit is 'count' and the volume unit can be one of ['L', 'dL', 'mL', 'μL', 'mm$^3$', 'cm$^3$', 'm$^3$']. Examples: {'106': 15034 count/μL}, {'63': 1750000 count/L]}. For all other numeric parameters whose labels are not in the form of concentrations (mass/volume), you MUST use one of the EXACT label names provided in the Values & Type column. Examples: {'66': 39 degrees celsius}. Please output a JSON dict in the mentioned format, specifying the value for each parameter ID in the provided table.

**Output**

```
{
    "2": "70 years",
    "13": "87 mm Hg",
    "14": "55 mm Hg",
    "93": "39 mg/dL",
    "102": "18 breaths per minute",
    "176": "Not Mentioned"
}
```

At the end of this step, we parsed the parameter extractions for each note to ensure that all numeric attributes had an integer or decimal value and categorical variables were either "Not Mentioned," or exactly matched one of the options provided in the Values & Units column. After manually verifying

the parameter extractions, we then had 627 instances which covered 34 calculators of our dataset. We capped a maximum of 20 notes for each calculator and so some calculators had less than 20 notes. The remaining 21 calculators had their notes and extracted parameters either produced using template-based functions implemented in Python (equation-based calculators) or the notes and the needed parameters were handwritten by clinicians (rule-based calculators).

## A.2 Templates for Synthetic Notes

For the following 11 calculators, we did not acquire any patient notes from Open-Patients. Instead, we used a template implemented in Python to create a patient note with the necessary values needed for each of the calculators:

1. QTc Calculators - {Bazett, Framingham, Rautaharju, Hodges, Fredericia}
2. Target Body Weight
3. MME Conversion
4. Steroid Conversion
5. Estimated Due Date
6. Estimated Gestational Age
7. Estimated Date of Conception

For each of the notes, we take random, but clinically plausible values for the required attributes needed for a particular calculator using the `random` library from Python. Shown below is an example of how we generate a note using a template for the MME Conversion calculator:

```
def mme_conversion():

    mme_drugs = ["Codeine", "FentaNYL␣buccal", "FentANYL␣patch", "
    HYDROcodone", "HYDROmorphone", "Methadone", "Morphine", "OxyCODONE", "
    OxyMORphone", "Tapentadol", "TraMADol"]

    drugs = random.sample(mme_drugs, 3)

    note = "The␣patient␣takes␣"

    input_parameters = {}

    for i in range(3):

        num_doses = random.randint(1, 3)
        num_amount = round(random.randint(1, 7)) * 10

        key_name_dose = drugs[i] + "␣Dose"
        key_name_dose_per_day = drugs[i] + "␣Dose␣Per␣Day"

        if drugs[i] == "FentaNYL␣buccal" or drugs[i] == "FentaNYL␣patch":
            input_parameters[key_name_dose] = [num_amount , "$\mu$g"]
        else:
            input_parameters[key_name_dose] = [num_amount , "mg"]

        input_parameters[key_name_dose_per_day] = [num_doses, "per␣day"]

        add_s = 's'

        if num_doses == 1:
            add_s = ''
```

```
        if i == len(drugs) - 1:
            note += f"and␣{num_amount}␣mg␣of␣{drugs[i]}␣{num_doses}␣time{
add_s}␣a␣day."
        else:
            note += f"{num_amount}␣mg␣of␣{drugs[i]}␣{num_doses}␣time{add_s}
␣a␣day,␣"

    return note, input_parameters
```

Hence, making calls to `mme_conversion()` would generate patient notes which all follow the same structure, but with different drugs and dosage amounts. Here are three such instances generated by the `mme_conversion()` function:

1. The patient takes 70 mg of OxyMORphone 3 times a day, 60 mg of Codeine 2 times a day, and 30 mg of FentaNYL buccal 1 time a day.

2. The patient takes 30 mg of TraMADol 2 times a day, 40 mg of OxyMORphone 1 time a day, and 50 mg of OxyCODONE 1 time a day.

3. The patient takes 60 mg of HYDROmorphone 1 time a day, 50 mg of Codeine 2 times a day, and 30 mg of Methadone 1 time a day.

We used a similar approach for the other 10 calculators for which we needed a synthesized template. These templates can be found in `synthesize_patient_note.py` file of the Github repository.

### A.3    Clinician Synthesized Notes

For the following 10 rule-based calculators, there were no notes curated from PMC-Patients and so we had notes synthesized from clinicians based on a set of pre-annotated values: Revised Score for Cardiac Risk index, HAS-BLED Score, Charlson Comorbidity Index, PSI Score, Child-Pugh Score for Cirrhosis Mortality, Glasgow Coma Score, APACHE II, SODA, Caprini Score for Venous Thromboembolism.

For synthesizing the notes, the following instructions were provided:

- **Calculator Information:** For each calculator, we provide a link from MDCalc.com which computes the medical value associated with a calculator, given the necessary inputs. You can use this link to get more information about the calculator, the details of the attributes and the values needed for a calculator, and any details about how a computation is performed.

- **How to synthesize a patient note:** Once you familiarize yourself with the calculator, please take a look at the Excel sheet provided for a given calculator. Here is the information about each of the columns:

  – The first column ("Variable Name") lists all of the attribute names needed for a given calculator.

  – The second column ("Values & Units") will correspond to the possible values or units that the attribute can take on. For attributes that are categorical and do not take an integer or decimal value, you **MUST** use one of the values provided specified in the "Values & Units" column for that given attribute (i.e. for the attribute "Hepatic Disease History" if the "Values & Units" column lists "True, False" as the possible attributes, then you must use one of these). If an attribute is numerical, then you MUST use an integer (i.e. 43 years for the attribute "age") or a decimal value (i.e. 3.32 mg/dL for the attribute "Creatinine").

  – The third column ("Pre-Annotated Value") will correspond to the assigned value for a given attribute for a specific patient note. These are the values that are initially assigned for each item, but you can change them if they are not at all humanly plausible. In this case, you should adhere to the rules mentioned in the second bullet point for assigning an appropriate value and units. As mentioned, you can skip some descriptive (non-numeric) attributes and in this case, the value should be "Not Mentioned."

  – For some calculators, there will be a fourth column ("Description") that will provide more context about the attribute. It may also specify the criteria that need to be met if someone assigns a particular value for a given attribute.

In all, you need to synthesize a patient note based on the attributes provided in the first column and assign these attributes to the values provided in the third column.

To make the patient note as authentic as possible, for attributes that take on categorical values, try to avoid using the same exact name of the value in your note.

For example, from the calculator, Charlson Comorbidity Index (CCI), one of the attributes checks for the presence of liver disease with the following values: (None, Mild, Moderate to Severe). Instead of saying "The patient's liver disease severity is classified as 'mild,'" write something like "The patient presents with symptoms suggestive of chronic hepatitis, including persistent fatigue, abdominal discomfort, and elevated liver enzymes, warranting further evaluation for viral hepatitis markers and liver function tests." By not directly specifying the value associated with liver disease, the patient note reads more authentically and we can use this to benchmark an LLM's ability to properly determine the severity of the patient's liver disease when provided with the categories.

Similarly, for the attribute "Moderate to Severe CKD" in the CCI calculator, instead of writing, "the patient did not pass the screening for chronic kidney disease," write something like "the patient's serum creatinine is 1.23 mg/dL." This is equally indicative that the "Moderate to Severe CKD" attribute in the CCI calculator should be false. For numerical attributes, you can directly specify the value, but you should try to vary the units associated with the value. This is not to say that you can never directly specify an attribute with a value provided from the third column, but you should vary your methods of conveying the values for an attribute so that we can measure how good an LLM is at deducing the value for a given attribute.

Based on these instructions, clinicians synthesized 20 notes for each of the 10 rule-based calculators. They also provided the extracted parameter values in the same dictionary format that was used for obtaining the ground truth parameter values from Open-Patients.

### A.4 Templates for Natural Language Explanation

A core component of MEDCALC-BENCH is the natural language explanations for showing how the final answer is obtained. For each of the 55 calculators, we implement a function that takes in the input parameters needed for the calculator and outputs a step-by-step natural language explanation of how the final answer is obtained.

It should be noted that the parameter extractions from Section A.1 needed to go through an additional parsing step before they could be passed into template-based explanation functions. Firstly, we mapped each extracted parameter ID to the input variable name that would be used in Python. Additionally, for numerical attributes such as creatinine concentration, age, weight, etc., we extracted the integer/decimal along with the label from the string. For categorical variables whose string value was either "True" or "False," we converted these strings to boolean literals. Lastly, for categorical variables that required a specific value based on the MDCalc list, we ensured the value given by GPT-4 matched one of the values provided by MDCalc.

Shown below is an example of this processing done for the input parameters for a patient note using the Glasgow-Blatchford Bleeding Score:

**Extracted Parameters:**

```
{
    "96": "False",
    "93": "34 mg/dL",
    "95": "False",
    "20": "80 beats per minute",
    "92": "13 g/dL",
    "1": "Female",
    "13": "90 mm Hg",
    "97": true,
    "94": true
}
```

From this, we transform the extracted attribute names to their names as Python variables for the template-based explanation function. We also convert the values into a format that can be used for the template-based explanation functions:

**Python Input Parameters:**

```
{
    "hepatic_disease_history": false,
    "bun": [
        34.0,
        "mg/dL"
    ],
    "syncope": false,
    "heart_rate": [
        80.0,
        "beats per minute"
    ],
    "hemoglobin": [
        13.0,
        "g/dL"
    ],
    "sex": "Female",
    "sys_bp": [
        90.0,
        "mm Hg"
    ],
    "cardiac_failure": true,
    "melena_present": true
}
```

It should be noted that not all categorical variables may be reported in the present in a patient note. Hence, if the value of the variable cannot be inferred, we state that it is not mentioned inside the template, and then report that we assume it to be false.

Based on this structure for generating the inputs to the explanation-based functions, we have implemented a template for providing a step-by-step explanation for all 55 calculators. Here is an example for how Glasgow-Blatchford Bleeding Score (GBS) explanation template was implemented:

```
def glasgow_bleeding_score_explanation(input_parameters):

    score = 0
```

```python
hemoglobin_exp, hemoglobin = unit_converter_new.conversion_explanation(
input_parameters["hemoglobin"][0], "hemoglobin", 64500, None, input_parameters[
"hemoglobin"][1], "g/dL")
bun_exp, bun = unit_converter_new.conversion_explanation(input_parameters["bun"
][0], "BUN", 28.08, None, input_parameters["bun"][1], "mg/dL")
gender = input_parameters["sex"]
systiolic_bp = input_parameters["sys_bp"][0]
heart_rate = input_parameters["heart_rate"][0]

explanation = f"The current glasgow bleeding score is 0. The patient's gender is
 {gender}.\n"
explanation += hemoglobin_exp

if gender == "Male":
    if 12 < hemoglobin <= 13:
        explanation += f"Because the patient is a male and the hemoglobin
concentration is between 12 and 13 g/dL, we add one point, making the current
score {score} + 1 = {score + 1}.\n"
        score += 1
    elif 10 <= hemoglobin < 12:
        explanation += f"Because the patient is a male and the hemoglobin
concentration is between 10 and 12 g/dL, we add three points, making the
current score {score} + 3 = {score + 3}.\n"
        score += 3
    elif hemoglobin < 10:
        explanation += f"Because the patient is a male and the hemoglobin
concentration is less than 10 and 12 g/dL, we add six points, making the
current score {score} + 6 = {score + 6}.\n"
        score += 6
    elif hemoglobin > 13:
        explanation += f"Because the patient is a male and the hemoglobin
concentration is greater than 13 g/dL, we do not add any points, keeping the
current score at {score}.\n"

else:
    if 10 < hemoglobin <= 12:
        explanation += f"Because the patient is a female and the hemoglobin
concentration is between 10 and 12 mg/dL, we add one point, making the current
score {score} + 1 = {score + 1}.\n"
        score += 1
    elif hemoglobin < 10:
        explanation += f"Because the patient is a female and the hemoglobin
concentration is less than 10 mg/dL, we add three points, making the current
score {score} + 3 = {score + 3}.\n"
        score += 6
    elif hemoglobin > 12:
        explanation += f"Because the patient is a female and the hemoglobin
concentration is greater than 12 mg/dL, we do not add any points, keeping the
current score at {score}.\n"

explanation += bun_exp

if 18.2 <= bun < 22.4:
    explanation += f"The BUN concentration is between 18.2 and 22.4 mg/dL, and
so we add two points, making the current score {score} + 2 = {score + 2}.\n"
    score += 2
elif 22.4 <= bun < 28:
    explanation += f"The BUN concentration is between 22.4 and 28 mg/dL, and so
we add three points, making the current score {score} + 3 = {score + 3}.\n"
    score += 3
elif 28 <= bun < 70:
    explanation += f"The BUN concentration is between 28 and 70 mg/dL, and so we
 add four points, making the current score {score} + 4 = {score + 4}.\n"
    score += 4
```

```python
elif bun > 70:
    explanation += f"The␣BUN␣concentration␣is␣greater␣than␣70␣mg/dL,␣and␣so␣we␣
add␣six␣points,␣making␣the␣current␣score␣{score}␣+␣6␣=␣{score␣+␣6}.\n"
    score += 6
elif bun < 18.2:
    explanation += f"The␣BUN␣concentration␣is␣less␣than␣18.2␣mg/dL,␣and␣so␣we␣do
␣not␣make␣any␣changes␣to␣the␣score,␣keeping␣the␣score␣at␣{score}.\n"

explanation += f"The␣patient's␣blood␣pressure␣is␣{systiolic_bp}␣mm␣Hg.␣"

if 100 <= systiolic_bp < 110:
    explanation += f"Because␣the␣patient's␣systolic␣blood␣pressure␣is␣between␣
100␣and␣110␣mm␣Hg,␣we␣increase␣the␣score␣by␣one␣point,␣making␣the␣current␣score
␣{score}␣+␣1␣=␣{score␣+␣1}.\n"
    score += 1
elif 90 <= systiolic_bp < 100:
    explanation += f"Because␣the␣patient's␣systolic␣blood␣pressure␣is␣between␣90
␣and␣100␣mm␣Hg,␣we␣increase␣the␣score␣by␣two␣points,␣making␣the␣current␣score␣{
score}␣+␣2␣=␣{score␣+␣2}.\n"
    score += 2
elif systiolic_bp < 90:
    explanation += f"Because␣the␣patient's␣systolic␣blood␣pressure␣is␣less␣than␣
90␣mm␣Hg,␣we␣increase␣the␣score␣by␣three␣points,␣making␣the␣current␣score␣{
score}␣+␣3␣=␣{score␣+␣3}.\n"
    score += 3
elif systiolic_bp >= 110:
    explanation += f"Because␣the␣patient's␣systolic␣blood␣pressure␣is␣greater␣
than␣or␣equal␣to␣110␣mm␣Hg,␣we␣do␣not␣add␣points␣to␣the␣score,␣keeping␣the␣
current␣score␣at␣{score}␣+␣3␣=␣{score␣+␣3}.\n"

explanation += f"The␣patient's␣heart␣rate␣is␣{heart_rate}␣beats␣per␣minute.␣"

if heart_rate >= 100:
    explanation += f"Because␣the␣heart␣rate␣is␣greater␣or␣equal␣to␣than␣100␣
beats␣per␣minute,␣we␣increase␣the␣score␣by␣one␣point,␣making␣the␣current␣score␣{
score}␣+␣1␣=␣{score␣+␣1}.\n"
    score += 1
else:
     explanation += f"Because␣the␣heart␣rate␣is␣less␣than␣100␣beats␣per␣minute,␣
we␣do␣not␣change␣the␣score,␣keeping␣the␣current␣score␣at␣{score}.\n"

default_parameters = {"melena_present": "melena", "syncope": "recent␣syncope", "
hepatic_disease_history": "hepatic␣disease␣history", "cardiac_failure": "
cardiac␣failure"}

for parameter in default_parameters:
    if parameter not in input_parameters:
        explanation += f"The␣patient's␣status␣for␣{default_parameters[parameter
]}␣is␣missing␣from␣the␣patient␣note␣and␣so␣we␣assume␣it␣is␣absent␣from␣the␣
patient.\n"
        input_parameters[parameter] = False
        explanation += f"Hence,␣we␣do␣not␣add␣any␣points␣to␣the␣score,␣keeping␣
it␣at␣{score}.\n"

    elif parameter in ['syncope', 'hepatic_disease_history', 'cardiac_failure']
and input_parameters[parameter]:
        explanation +=  f"The␣patient␣has␣a␣{default_parameters[parameter]},␣and
␣so␣we␣add␣two␣points␣to␣the␣current␣total,␣making␣the␣current␣total␣{score}␣+␣
2␣=␣␣{score␣+␣2}.\n"
        score += 2

    elif input_parameters[parameter]:
```

```
            explanation += f"The␣patient␣has␣{default_parameters[parameter]}␣and␣so
␣we␣add␣one␣point␣to␣the␣current␣total,␣making␣the␣current␣total␣{score}␣+␣1␣=␣
␣{score␣+␣1}.\n"
            score += 1

        else:
            explanation += f"The␣patient's␣status␣for␣{default_parameters[parameter
]}␣is␣reported␣to␣be␣absent␣for␣the␣patient,␣and␣so␣we␣do␣not␣add␣any␣points,␣
keeping␣the␣current␣total␣at␣{score}.\n"

    explanation += f"The␣patient's␣Glasgow␣Bleeding␣Score␣is␣{score}.\n"

    return {"Explanation": explanation, "Answer": score, "Calculator␣Answer":
    glasgow_bleeding_score(input_parameters)}
```

After applying the GBS explanation functionglasgow_bleeding_score_explanation, onto the python input parameters provided in Section A.4, we get the following output explanation:

```
The current Glasgow bleeding score is 0.  The patient's gender is Female.
The concentration of hemoglobin is 13.0 g/dL. Because the patient is a
female and the hemoglobin concentration is greater than 12 mg/dL, we do
not add any points, keeping the current score at 0.  The concentration of
BUN is 34.0 mg/dL. The BUN concentration is between 28 and 70 mg/dL, and so
we add four points, making the current score 0 + 4 = 4.The patient's blood
pressure is 90.0 mm Hg.  Because the patient's systolic blood pressure is
between 90 and 100 mm Hg, we increase the score by two points, making the
current score 4 + 2 = 6.The patient's heart rate is 80.0 beats per minute.
Because the heart rate is less than 100 beats per minute, we do not change
the score, keeping the current score at 6.The patient has melena and so
we add one point to the current total, making the current total 6 + 1 =
7.  The patient's status for recent syncope is reported to be absent for
the patient, and so we do not add any points, keeping the current total at
7.The patient's status for hepatic disease history is reported to be absent
for the patient, and so we do not add any points, keeping the current total
at 7.The patient has a cardiac failure, and so we add two points to the
current total, making the current total 7 + 2 = 9.  The patient's Glasgow
Bleeding Score is 9.
```

### A.5 MEDCALC-BENCH Calculators Covered

MEDCALC-BENCH involves 55 different calculators. Shown below is a table providing more information about each calculator:

Table 4: Information for each calculator in MEDCALC-BENCH test set

| Calculator Name | Type | Sub-Type | Number of Attributes | Number of Notes |
|---|---|---|---|---|
| Creatinine Clearance (Cockcroft-Gault Equation) | Equation-Based | Lab Test | 5 | 20 |
| CKD-EPI Equations for Glomerular Filtration Rate | Equation-Based | Lab Test | 3 | 20 |
| CHA2DS2-VASc Score for Atrial Fibrillation Stroke Risk | Rule-Based | Risk | 10 | 20 |
| Mean Arterial Pressure (MAP) | Equation-Based | Physical | 3 | 20 |
| Body Mass Index (BMI) | Equation-Based | Physical | 3 | 20 |
| Calcium Correction for Hypoalbuminemia | Equation-Based | Lab Test | 2 | 20 |

*Continued on next page*

Table 4: Information for each calculator in MEDCALC-BENCH test dataset (continued)

| Calculator Name | Type | Sub-Type | Number of Attributes | Number of Notes |
|---|---|---|---|---|
| Wells' Criteria for Pulmonary Embolism | Rule-Based | Risk | 9 | 20 |
| MDRD GFR Equation | Equation-Based | Lab Test | 4 | 20 |
| Ideal Body Weight | Equation-Based | Physical | 2 | 20 |
| QTc Bazett Calculator | Equation-Based | Physical | 2 | 20 |
| Child-Pugh Score for Cirrhosis Mortality | Rule-Based | Severity | 5 | 20 |
| Wells' Criteria for DVT | Rule-Based | Risk | 11 | 20 |
| Revised Cardiac Risk Index for Pre-Operative Risk | Rule-Based | Risk | 6 | 20 |
| HEART Score for Major Cardiac Events | Rule-Based | Risk | 13 | 20 |
| Fibrosis-4 (FIB-4) Index for Liver Fibrosis | Equation-Based | Lab Test | 4 | 20 |
| Centor Score (Modified/McIsaac) for Strep Pharyngitis | Rule-Based | Severity | 5 | 20 |
| Maintenance Fluids Calculations | Equation-Based | Physical | 1 | 20 |
| MELD Na (UNOS/OPTN) | Equation-Based | Lab Test | 6 | 5 |
| HAS-BLED Score for Major Bleeding Risk | Rule-Based | Risk | 10 | 20 |
| Sodium Correction for Hyperglycemia | Equation-Based | Lab Test | 2 | 20 |
| Glasgow-Blatchford Bleeding Score (GBS) | Rule-Based | Risk | 9 | 20 |
| Serum Osmolality | Equation-Based | Lab Test | 3 | 20 |
| HOMA-IR (Homeostatic Model Assessment for Insulin Resistance) | Equation-Based | Lab Test | 2 | 2 |
| Charlson Comorbidity Index (CCI) | Rule-Based | Risk | 18 | 20 |
| FeverPAIN Score for Strep Pharyngitis | Rule-Based | Diagnosis | 5 | 20 |
| Free Water Deficit | Equation-Based | Lab Test | 4 | 20 |
| Anion Gap | Equation-Based | Lab Test | 3 | 20 |
| Fractional Excretion of Sodium (FENa) | Equation-Based | Lab Test | 4 | 5 |
| LDL Calculated | Equation-Based | Lab Test | 3 | 20 |
| CURB-65 Score for Pneumonia Severity | Rule-Based | Risk | 6 | 20 |
| Framingham Risk Score for Hard Coronary Heart Disease | Equation-Based | Lab Test | 7 | 16 |
| PERC Rule for Pulmonary Embolism | Rule-Based | Diagnosis | 9 | 20 |
| SIRS Criteria | Rule-Based | Diagnosis | 5 | 20 |
| QTc Fridericia Calculator | Equation-Based | Physical | 2 | 20 |
| QTc Framingham Calculator | Equation-Based | Physical | 2 | 20 |
| QTc Hodges Calculator | Equation-Based | Physical | 2 | 20 |
| QTc Rautaharju Calculator | Equation-Based | Physical | 2 | 20 |
| Body Surface Area Calculator | Equation-Based | Physical | 2 | 20 |
| Adjusted Body Weight | Equation-Based | Physical | 3 | 20 |
| Delta Gap | Equation-Based | Lab Test | 3 | 20 |
| Delta Ratio | Equation-Based | Lab Test | 3 | 20 |
| Albumin Corrected Anion Gap | Equation-Based | Lab Test | 4 | 20 |
| Albumin Corrected Delta Gap | Equation-Based | Lab Test | 4 | 20 |

*Continued on next page*

Table 4: Information for each calculator in MEDCALC-BENCH test dataset (continued)

| Calculator Name | Type | Sub-Type | Number of Attributes | Number of Notes |
|---|---|---|---|---|
| Albumin Corrected Delta Ratio | Equation-Based | Lab Test | 4 | 20 |
| PSI Score: Pneumonia Severity Index for CAP | Rule-Based | Severity | 20 | 20 |
| Glasgow Coma Score (GCS) | Rule-Based | Severity | 3 | 20 |
| APACHE II Score | Rule-Based | Risk | 20 | 20 |
| Sequential Organ Failure Assessment (SOFA) Score | Rule-Based | Risk | 16 | 20 |
| Caprini Score for Venous Thromboembolism (2005) | Rule-Based | Risk | 31 | 20 |
| Estimated Due Date | Equation-Based | Date | 1 | 20 |
| Steroid Conversion Calculator | Equation-Based | Dosage | 11 | 20 |
| Target weight | Equation-Based | Physical | 2 | 20 |
| Morphine Milligram Equivalents (MME) Calculator | Equation-Based | Dosage | 14 | 20 |
| Estimated Date of Conception | Equation-Based | Date | 1 | 20 |
| Estimated Gestational Age | Equation-Based | Date | 2 | 20 |

In addition to the 1,047 instances for MEDCALC-BENCH, we also curated a training dataset of 10,053 to fine-tune open-source LLMs. We followed the exact same procedure as the one we used for the testing dataset, except authors of this paper did not manually verify the parameter extractions from GPT-4. In spite of this, we still gained a significant increase in performance for the two open-source LLMs that we fine-tuned on using this dataset. More details on the training dataset can be found in Section C.

## B  Dataset License and Usage

### B.1  Dataset License

As mentioned in Section A.1, the notes come from Open-Patients, templates, or are handwritten by clinicians. The notes coming from the latter two are our property, but the notes from Open-Patients were created using existing datasets (TREC, MedQA, and PMC-Patients). We have verified that we can use these datasets for making MEDCALC-BENCH.

Specifically, although there is no issued license for the TREC datasets, the clinical trials and clinical decision support datasets from TREC are government-released datasets that were released for public use and distribution. The MedQA data is released by the MIT License. Hence, we can use instances from this dataset for MEDCALC-BENCH. Lastly, the PMC-Patients dataset is released by the CC-BY-SA 4.0 license. This gives us permission to re-distribute the dataset for MEDCALC-BENCH as long as it is not for commercial use. Additionally, because the PMC-Patients dataset is released by CC-BY-SA 4.0 license, we had to release Open-Patients, and consequently MEDCALC-BENCH, by the same license. Hence, both the training and testing instances of the MEDCALC-BENCH dataset are released by the **CC-BY-SA 4.0 license** and can be distributed for any non-commercial purposes.

The dataset and all the code for reproducing our results can be found at: `https://github.com/ncbi-nlp/MedCalc-Bench`. Additionally, we provide access to the training and test sets for MEDCALC-BENCH in a Croissant format using HuggingFace dataset as well: `https://huggingface.co/datasets/ncbi/MedCalc-Bench-v1.0`.

## B.2   Dataset Instance Metadata

For the MEDCALC-BENCH dataset uploaded on HuggingFace, each instance in both the training and testing dataset of MEDCALC-BENCH is a row that contains the following information:

- **Row Number** - Specifies the index of the instance.
- **Calculator ID** - Specifies the integer ID of the calculator.
- **Calculator Name** - Specifies the name of the clinical calculation task.
- **Category** - Specifies the sub-category of the calculator. For equation-based calculators, the options are lab test, dosage, date, or physical and for rule-based calculators, the options are risk, severity, and diagnosis.
- **Output Type** - Specifies the format type that the calculator will return. The options are decimal, integer, date (MM/DD/YY), or time in terms of weeks and days (i.e. (17 weeks, 4 days)).
- **Note ID** - Specifies the ID of the patient note. The ID of the note will either be the ID given by Open-Patients or it will be an integer value if the patient note was handwritten by clinicians or synthesized by a template.
- **Patient Note** - Specifies the patient note which provides the information needed to compute the final answer.
- **Question** - Specifies the question that is asked to the model to compute a specific medical value based on a particular calculator.
- **Relevant Entities** - Provides a dictionary of the parameters and their extracted values based on the patient note.
- **Ground Truth Answer** - Specifies the ground truth value without any units for the medical value that needs to be calculated.
- **Lower Limit** - For equation-based calculators whose output is a decimal, this value is 95% of the ground truth answer value. For all other cases, the lower limit is the same as the ground-truth value.
- **Upper Limit** - For equation-based calculators whose output is a decimal, this is value is 105% of the ground truth answer value. For all other cases, the upper limit is the same as the ground-truth value.
- **Ground Truth Explanation** - A paragraph for the data instance providing a step-by-step explanation for how the final answer was obtained.

## B.3   Reproducing Results

The main results for our dataset are the performances of various LLMs under different prompt settings. These are all shown in Table 2. First, please create a conda environment and install packages in the `requirements.txt` file. Then, please add your OpenAI API key to this environment. To obtain these results, simply execute the following command: `python run.py - model <model_name> - prompt <prompt_style>`.

The 8 options for the `model` argument as follows:

- Mistral 7B: mistralai/Mistral-7B-Instruct-v0.2
- Mixtral 8x7B: mistralai/Mixtral-8x7B-Instruct-v0.1
- Llama3-8B: meta-llama/Meta-Llama-3-8B-Instruct
- Llama3-70B: meta-llama/Meta-Llama-3-70B-Instruct
- Meditron-70B: epfl-llm/meditron-70b
- PMC-Llama-13B: axiong/PMC_LLaMA_13B

- GPT-3.5: OpenAI/gpt-3.5-turbo
- GPT-4: OpenAI/gpt-4

Additionally, we provide three options for the `prompt` argument:

- Direct answer = `direct_answer`
- Zero shot = `zero_shot`
- One shot = `one_shot_cot`

All open-source LLMs are run on 4 A100-80B GPUs for model inference. We provide the settings for the first three prompts settings below:

Table 5: Number of tokens used for zero-shot direct prompting

| PMC-LLaMA | MEDITRON | Mistral | Mixtral | Llama 3-8B | Llama 3-70B | GPT-3.5 | GPT-4 |
|---|---|---|---|---|---|---|---|
| Input | 276.9k | 736.9k | 669.9k | 669.9k | 727.5k | 727.5k | 662.5k | 662.5k |
| Output | 23 | 10.6k | 130.1k | 36.7k | 8.6k | 7.4k | 8.1k | 8.5k |

Table 6: Number of tokens used for zero-shot CoT prompting

| | PMC-LLaMA | MEDITRON | Mistral | Mixtral | Llama 3-8B | Llama 3-70B | GPT-3.5 | GPT-4 |
|---|---|---|---|---|---|---|---|---|
| Input | 332.6k | 823.8k | 721.2k | 721.2k | 778.8k | 778.8k | 713.8k | 713.8k |
| Output | 240.9k | 18.0k | 241.5k | 222.6k | 230.3k | 223.9k | 191.4k | 216.8k |

Table 7: Number of tokens used for one-shot CoT prompting

| | PMC-LLaMA | MEDITRON | Mistral | Mixtral | Llama 3-8B | Llama 3-70B | GPT-3.5 | GPT-4 |
|---|---|---|---|---|---|---|---|---|
| Input | 861.3k | 1.4M | 1.8M | 1.8M | 1.8M | 1.8M | 1.8M | 1.8M |
| Output | 222.7k | 387.7k | 263.9k | 290.3k | 360.4k | 351.7k | 327.7k | 330.4k |

Upon executing `run.py`, the results will be saved in a file called `<model>_<prompt>.jsonl`. Each instance will have the following metadata associated with them:

```
{
  "Row Number": ,
  "Calculator Name": ,
  "Calculator ID": ,
  "Category": ,
  "Note ID": ,
  "Question": ,
  "LLM Answer": ,
  "LLM Explanation": ,
  "Ground Truth Answer": ,
  "Ground Truth Explanation": ,
  "Result":
}
```

Here is what each item means:

- **"Row Number"** - specifies the row in the MEDCALC-BENCH CSV
- **"Calculator Name"** - specifies which calculator that is being covered by this instance
- **"Calculator ID"** - unique ID of the calculator
- **"Category"** - sub-category for the calculator

- **"Note ID"** - specifies the Note ID from MEDCALC-BENCH
- **"Patient Note"** - provides the patient note covered by this instance
- **"Question"** - question being asked by this instance
- **"LLM Answer"** - the final answer value given by the LLM
- **"LLM Explanation"** - the explanation provided by the LLM for the problem
- **"Ground Truth Answer"** - the ground truth answer for the instance
- **"Ground Truth Explanation"** - the ground truth explanation for the instance
- **"Result"** - either "Correct" or "Incorrect" by comparing the LLM Answer and Ground Truth Answer

Note that in the direct answer setting, the LLM is only expected to provide a direct answer and so the **"LLM Explanation"** section will be "N/A" for all of them.

### B.4    Author Statement

Although we have verified that our dataset is available by CC-BY-SA 4.0 license, all the authors bear responsibility for any infringements. For any updates on the dataset that needs to be made (i.e. adding new calculators, more notes, ect), we will update the Github repository and still keep archives of previous versions of the dataset on both the repository and HuggingFace.

## C    Training Dataset for Fine-Tuning on MEDCALC-BENCH

### C.1    Training Dataset

In addition to the 1,047 manually verified instances for MEDCALC-BENCH, we also curated a training dataset of 10,053 instances consisting of patient notes, questions, final answers, and explanations using the same extraction process that we used for curating the test set. This training dataset contained notes for 40 calculators. These calculators either had over 20 patient notes from Open-Patients or were one of the 11 calculators which had their notes synthesized from a template-based function (80 notes were synthesized for these calculators). The remaining 15 calculators had no patient notes to train with inside the training dataset. Shown in the table below are the number of instances for each calculator in the training dataset, along with their calculator type and sub-type.

Table 8: Information about each calculator in MEDCALC-BENCH training dataset

| Calculator Name | Type | Sub-Type | Number of Attributes | Number of Notes |
|---|---|---|---|---|
| Creatinine Clearance (Cockcroft-Gault Equation) | Equation-Based | Lab Test | 5 | 157 |
| CKD-EPI Equations for Glomerular Filtration Rate | Equation-Based | Lab Test | 3 | 519 |
| CHA2DS2-VASc Score for Atrial Fibrillation Stroke Risk | Rule-Based | Risk | 10 | 517 |
| Mean Arterial Pressure (MAP) | Equation-Based | Physical | 3 | 948 |
| Body Mass Index (BMI) | Equation-Based | Physical | 3 | 519 |
| Calcium Correction for Hypoalbuminemia | Equation-Based | Lab Test | 2 | 212 |
| Wells' Criteria for Pulmonary Embolism | Rule-Based | Risk | 9 | 445 |
| MDRD GFR Equation | Equation-Based | Lab Test | 4 | 312 |
| Ideal Body Weight | Equation-Based | Physical | 2 | 621 |

*Continued on next page*

Table 8: Information about each calculator in MEDCALC-BENCH training dataset (continued)

| Calculator Name | Type | Sub-Type | Number of Attributes | Number of Notes |
|---|---|---|---|---|
| QTc Bazett Calculator | Equation-Based | Physical | 2 | 80 |
| Wells' Criteria for DVT | Rule-Based | Risk | 11 | 578 |
| HEART Score for Major Cardiac Events | Rule-Based | Risk | 13 | 222 |
| Fibrosis-4 (FIB-4) Index for Liver Fibrosis | Equation-Based | Lab Test | 4 | 120 |
| Maintenance Fluids Calculations | Equation-Based | Physical | 1 | 889 |
| Sodium Correction for Hyperglycemia | Equation-Based | Lab Test | 2 | 295 |
| Serum Osmolality | Equation-Based | Lab Test | 3 | 370 |
| FeverPAIN Score for Strep Pharyngitis | Rule-Based | Diagnosis | 5 | 42 |
| Free Water Deficit | Equation-Based | Lab Test | 4 | 296 |
| Anion Gap | Equation-Based | Lab Test | 3 | 216 |
| LDL Calculated | Equation-Based | Lab Test | 3 | 66 |
| CURB-65 Score for Pneumonia Severity | Rule-Based | Risk | 6 | 125 |
| PERC Rule for Pulmonary Embolism | Rule-Based | Diagnosis | 9 | 153 |
| SIRS Criteria | Rule-Based | Diagnosis | 5 | 252 |
| QTc Fridericia Calculator | Equation-Based | Physical | 2 | 80 |
| QTc Framingham Calculator | Equation-Based | Physical | 2 | 80 |
| QTc Hodges Calculator | Equation-Based | Physical | 2 | 80 |
| QTc Rautaharju Calculator | Equation-Based | Physical | 2 | 80 |
| Body Surface Area Calculator | Equation-Based | Physical | 2 | 929 |
| Target weight | Equation-Based | Physical | 2 | 80 |
| Adjusted Body Weight | Equation-Based | Physical | 3 | 570 |
| Delta Gap | Equation-Based | Lab Test | 3 | 331 |
| Delta Ratio | Equation-Based | Lab Test | 3 | 288 |
| Albumin Corrected Anion Gap | Equation-Based | Lab Test | 4 | 91 |
| Albumin Corrected Delta Gap | Equation-Based | Lab Test | 4 | 75 |
| Albumin Corrected Delta Ratio | Equation-Based | Lab Test | 4 | 74 |
| MME Conversion | Equation-Based | Dosage | 2 | 80 |
| Steroid Conversion | Equation-Based | Dosage | 2 | 80 |
| Estimated Due Date | Equation-Based | Date | 2 | 80 |
| Estimated Date of Conception | Equation-Based | Date | 2 | 80 |
| Estimated Gestational Age | Equation-Based | Date | 2 | 80 |

## C.2 Training Details

We fine-tuned Mistral-7B and Llama2-7B on our dataset. All fine-tuning runs are performed on a single 4xA100 40GB node. We use an adapted version of the Code-Act [?] training scripts which are based on a fork of Megatron-LLM [?]. We train Llama2-7b and not Llama3-7B which was benchmarked in the main paper as this model is not currently supported by Megatron-LLM. Both training and test data are converted into a chatML form. Shown below is an example:

**System Message**

Below is a patient note as well as a medical question about the patient. Provide an accurate answer to the question based on the note. Explain your reasoning before stating your final answer and put your final answer at the end of your response in the format Answer: INSERT_ANSWER

**User Message**

Patient Note:

A 78-year-old male with history of hypertension, dyslipidemia, and prosthetic aortic valve replacement presented to the emergency department with complaints of exertional shortness of breath for few days. Chest X-ray showed widened mediastinum. Subsequently, CT angiogram of the chest showed massively dilated ascending aorta with maximum diameter of 10.2 cm []. Transthoracic echocardiogram revealed normal ejection fraction with mild aortic insufficiency, well-functioning prosthetic aortic valve, and no aortic stenosis. His creatinine was 1.8 mg/dl, and he had received 150 ml of contrast for CT angiogram. It was necessary to evaluate his coronary arteries before surgery for AAA. There were two options in this case: coronary CT angiography or conventional coronary angiography. Coronary CT angiography would require about 150 ml of contrast.[] A diagnostic coronary angiogram can be done using about 50 ml of contrast.[] Given his renal insufficiency, we decided to do conventional coronary angiography. Since the ascending aorta and root were extremely dilated, we knew it would not be possible to engage the coronaries using regular catheters and standard technique. Hence, telescopic technique was used. Amplatz left 3 (AL3) 7F (French) guide catheter (90 cm) was initially used, and nonselective injection of contrast was done to see the coronary ostium [ and ]. After that, a 5F multipurpose (MP) catheter (110 cm) was telescoped through 7F AL3 guide catheter to engage the ostium of the left main coronary artery [ and ]. Using this technique, images of coronaries were obtained, and it showed minimal luminal irregularities in major epicardial coronary arteries. Total of 50 ml contrast was used. The patient underwent successful surgery with aortic valve replacement and excision of aneurysm with graft placement. The most likely etiology of giant aortic aneurysm in our patient was atherosclerosis. However, we did not have any record of his previous aortic valve surgery, and we do not know if it was a bicuspid aortic valve. In that case, he could have had aortopathy associated with bicuspid aortic valve.

**Question**: Using the MDRD GFR equation, what is the patient's Glomerular Filtration Rate (GFR) in terms of mL/min/1.73 m²?

**Response**:

The patient is 78 years old. The concentration of Creatinine is 1.8 mg/dL. The race of the patient is not provided, so the default value of the race coefficient is 1.0. The patient is male, so the gender coefficient is 1.0. The patient's estimated GFR is calculated using the MDRD equation as: The formula for GFR is given by: $175 \times \text{creatinine}^{-1.154} \times \text{age}^{-0.203} \times \text{race\_coefficient} \times \text{gender\_coefficient}$. The creatinine concentration is mg/dL. Plugging in these values will give us: $175 \times 1.8^{-1.154} \times 78^{-0.203} \times 1.0 \times 1.0 = 36.674$. Hence, the patient's GFR is 36.674 mL/min/1.73m².

Answer: 36.674

Following the implementation used for finetuning LLMs for CodeAct , we only calculate and optimize the loss on the assistant response, pack short instances into longer ones, and train using flash attention. We train for two epochs with a batch size of 32 on 4 parallel GPU's with a learning rate of 1e-5, 50 warmup steps, and cosine decay.

## C.3   Results

From the above fine-tuning procedure, we achieved the following results on the MEDCALC-BENCH test set:

Table 9: Results after fine-tuning open-source LLMs on MEDCALC-BENCH training dataset

| Model | Original Accuracy | Fine-tuned Accuracy |
|---|---|---|
| Mistral-7B | 10.79% | 49.19% |
| Llama-2-7B | 1.53% | 45.75% |

Both fine-tuning results significantly increased their respective models' performance with the lower Llama-2 performance likely due to its lower context length. Hence, we show the efficacy of the training dataset for MEDCALC-BENCH in improving a model's ability to recall the correct equations,

retrieve relevant parameters, and correctly perform arithmetic. Although these results demonstrate a significant performance improvement that fine-tuning has on our dataset, this still shows that more work needs to be done in improving LLMs for them to be reliable clinical calculators.

# D Code-Augmented LLMs Prompt Setting

## D.1 Code-Augmented LLM Performance

To minimize a model's arithmetic errors on MEDCALC-BENCH, we instead prompt a model to write code for performing any arithmetic operations. The user will then execute the code and provide the output from the console. If there are any compiling issues, the model has up to 20 tries to output the answer. For further details on this implementation, please refer to the `generate_code_prompt.py` file in the repository.

Due to limited compute, we only ran the code execution prompts for GPT-3.5 and GPT-4 and did not perform this experiment for open-source LLMs. Hence, we only included these results in the supplemental section and not the main paper as not all LLMs were benchmarked in this setting.

Shown below are the results for zero-shot chain-of-thought (CoT) for GPT-3.5-turbo and GPT-4, along with the code-augmented results for GPT-3.5-turbo-16k and GPT-4:

Table 10: Comparison of GPT-3.5-turbo and GPT-4 in zero-shot CoT before and after adding code interpreter

| | GPT-3.5-turbo | | GPT-4 | |
|---|---|---|---|---|
| | **Zero-Shot CoT** | **Code Prompt** | **Zero-Shot CoT** | **Code Prompt** |
| **Equation** | | | | |
| Lab Test | 20.49% | 30.04% | 26.30% | 49.85% |
| Physical | 45.00% | 56.76% | 71.25% | 81.66% |
| Dosage | 17.50% | 15.79% | 40.00% | 32.50% |
| Date | 11.67% | 28.33% | 48.33% | 43.33% |
| **Rule** | | | | |
| Severity | 10.00% | 13.33% | 15.00% | 18.75% |
| Diagnosis | 31.67% | 25.00% | 28.33% | 23.33% |
| Risk | 13.33% | 12.68% | 27.50% | 33.75% |
| **Overall** | 23.69% | 30.29% | 37.92% | 48.51% |

After augmenting with a code interpreter, the accuracies of GPT-3.5-turbo and GPT-4 have increased by 6.60% and 10.59%, respectively. Hence, we conclude that LLMs can significantly benefit from using a code interpreter to reduce its arithmetic mistakes and, thereby, become better at solving clinical calculations.

# E  Additional Analysis

## E.1  Accuracy vs. Number of Attributes

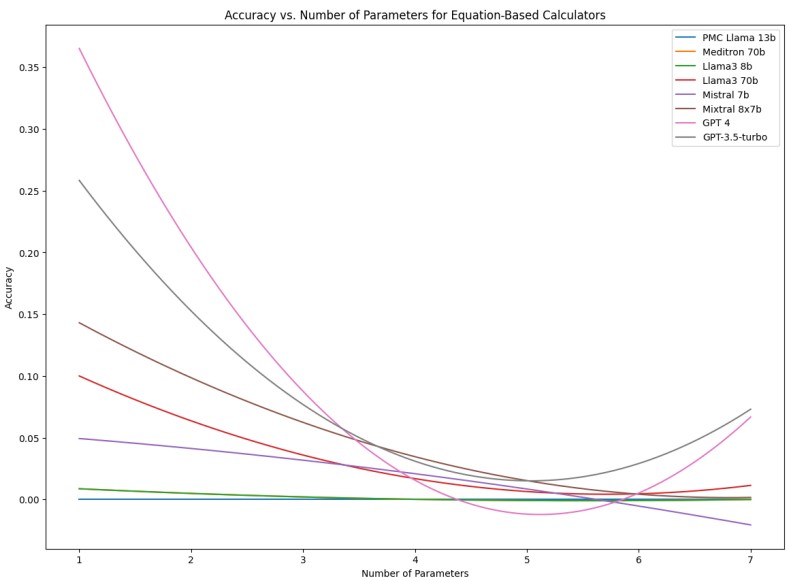

Figure 1: Accuracy vs. Number of Parameters for Equation-Based Calculators

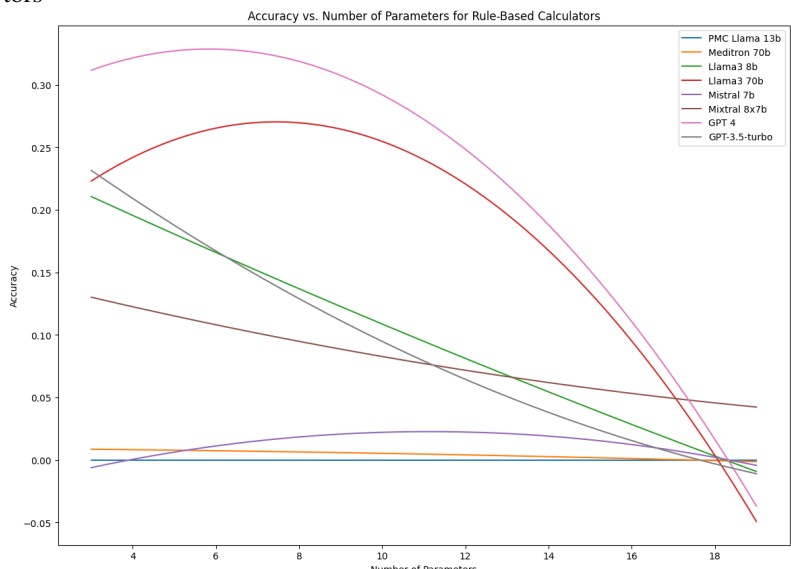

Figure 2: Accuracy vs. Number of Parameters for Rule-Based Calculators

While we have conducted analyses for LLM performance based on sub-category type, we also examine the performance for models based on the number of parameters. Using a line-of-best-fit for a second degree polynomial, we see the following trend for the number of parameters vs. accuracy for each of the eight models. Figure 1 shows the results for equation-based calculators and Figure 2 shows the results for rule-based calculators. As seen, the overall trend is that model performance tends to drop as the number of attributes increases. For models such as Meditron-70B and PMC-Llama-13B the drop is not as drastic, but that is because the accuracy was already close to 0 even for calculation tasks with the minimum number of attributes. However, for models such as Llama3-70B and GPT-4 which perform well relative to the other LLMs, the drop is much more drastic as the parameter

count increases. Hence, these plots show that even the highest-performing LLMs are at best able to make progress only on basic computation tasks and struggle significantly as the number of attributes increase.