# OpenReview forum: "MedCalc-Bench: Evaluating Large Language Models for Medical Calculations"
_NeurIPS.cc/2024/Datasets_and_Benchmarks_Track — NeurIPS 2024 Track Datasets and Benchmarks Oral_

### Official Review · Reviewer_aJHL · 2024-06-23
**Review of contribution 2157**

**Rating:** 8
**Confidence:** 3

**Review:**

(1) Large database for a very interesting and in the future relevant use case of LLMs so far not available. (2) Authors have created a very comprehensive benchmark of multiple LLMs using their dataset. (3) Paper shows the current limitations of LLMs for this use case and offers opportunities for improvement.

**Strengths:**

The availability of a large database for medical computation LLMs and calculation tasks highlights the complexity and necessity of the task. The poor performances of current models emphasize the need for ongoing research in this field. As a result, it remains an open challenge for the machine learning community.

**Additional Feedback:**

It is a good paper, focusing on a very interesting and most likely widely used application of LLMs in the future. Good figures and tables in the paper enhance the readability.
Room for improvement:
Table 1 – please add legend for L., I suppose it should read length?
Table 2 – is very busy with two decimals which suggest a non-existing level of accuracy, consider rounding to 1 decimal or even integers.
Table 4 – even busier, consider rounding to 1 decimal. Please add explanation to error types as a legend for quick cross referencing.

**Clarity:**

In general, well written paper. In some cases, the sentences are still a bit verbose and could be shortened.
Exp.: In my opinion lines 42-60 are duplicative to the text in chapter 2, although I liked the bullet points. I acknowledge by being a physician that medical writing style is different than the style of the engineering sciences.

Also paragraphs 5.2 and 5.3 are oddly sticking out. Please either drop them entirely, or elaborate more. If you decide to keep them, I don’t see the point in separating them. As such, 5.3 could be integrated into 5.2.

**Correctness:**

Claims made in the submission seem correct, and have important implications for other researchers. Benchmark experiments are well designed and have very nice and reproducible code accompanying them.

**Documentation:**

There are sufficient details available on data collection and organization, and maintenance, and ethical and responsible use. Data link is working fine. Well done Github repository.

**Ethics:**

I found no ethical concerns.

**Limitations:**

Limitation section is very brief. Authors could comment a bit on way how to A) further improve the dataset B) actions to improve the computational abilities of LLMs.

**Opportunities For Improvement:**

Somewhere around line 98/99 I am missing information on the medical professionals assessing the cases: who were they? What was their agreement (e.g. interrater kappa)? Perhaps the Equator Prisma guidelines for systematic reviews might provide helpful guidance https://www.equator-network.org/reporting-guidelines/good-research-practices-for-comparative-effectiveness-research-defining-reporting-and-interpreting-nonrandomized-studies-of-treatment-effects-using-secondary-data-sources-the-ispor-good-research-pr/).

**Relation To Prior Work:**

The proposed database is compared with other existing database for the same or similar problems and it’s novelty is clearly stated. It also discusses related work.

**Summary And Contributions:**

This paper proposes an annotated dataset of 1,000 medical cases suitable for evaluating the performance of a LLM on equation-based and rule-based medical calculation tasks on 7 topics along with baseline results. The dataset has 55 tasks (equation-based: lab, date, physical, dosage; rule-based: risk, severity, diagnosis) with 40 to 327 instances/cases. Labelling was done by an unknown number of medical professionals. The cases were obtained from various publicly available sources not infringing any patients’ confidentiality.

---

> ### Author Rebuttal · Authors · 2024-08-14
>
> We thank the reviewer for their constructive feedback and positive comments. To address your concerns:
>
> > Somewhere around line 98/99 I am missing information on the medical professionals assessing the cases: who were they? What was their agreement (e.g. interrater kappa)? Perhaps the Equator Prisma guidelines for systematic reviews might provide helpful guidance https://www.equator-network.org/reporting-guidelines/good-research-practices-for-comparative-effectiveness-research-defining-reporting-and-interpreting-nonrandomized-studies-of-treatment-effects-using-secondary-data-sources-the-ispor-good-research-pr/).
>
> We employed senior medical students to verify the extracted attribute values when curating the dataset. We will change “medical professionals” to “annotators with medical background”. Specifically, their directions were to verify that the GPT-4 extracted attribute values for a given calculator were correct based on the patient’s health status. For example, we received assistance with verifying attributes such as whether the patient has renal disease in the note for the computation of the HAS-BLED score. Due to the limited number of annotators and the labor-intensive nature of the annotation task, each instance was only verified by one annotator. We will acknowledge this in the limitation section.
>
> > Limitation section is very brief. Authors could comment a bit on way how to A) further improve the dataset B) actions to improve the computational abilities of LLMs.
>
> Thank you for your suggestion. We will expand the “Limitations and Future Work” section to address both of these important questions.
>
> Specifically, for A), we can improve the dataset by adding more instances and calculators for each instance as there are currently only 1047 patient notes available for the test set, although this is comparable to commonly used biomedical benchmarks such as MedQA (1273 test instances) and PubMedQA (500 test instances). In particular, we hope to expand our dataset by having more calculators related to specific risk calculations and also having a broader range of parameters for equation-based calculators as the majority of equation-based calculators have between 3-5 parameters.
>
> Regarding B), we would like to mention that we have investigated two actions that could improve the computational abilities of LLMs. Firstly, we investigated whether having the LLM write code and automatically compile the code can improve the performance of LLMs in section C of the appendix. Under this code-augmented setting, we observed an improvement of 6.60% (from 23.69% to 30.29%) for GPT-3.5-turbo and an improvement of 10.59% (from 37.92% to 48.51%) for GPT-4. We have also investigated fine-tuning on a training set and presented the improved results for two open-source LLMs in section D of the appendix. We found a 38.40% improvement for Mistral-7B (from 10.79% to 49.19%) and a 44.22% improvement for Llama2-7B (1.53% to 45.75%). For more discussion, we will add in our limitations section that there are additional methods that can improve LLM’s ability to do computation that we should benchmark with for future work. This includes augmenting an LLM with an already implemented function of the calculator (https://www.medrxiv.org/content/10.1101/2023.12.13.23299881v1), training an LLM with step-by-step PPO (https://arxiv.org/abs/2312.08935) which has improved effects on GSM8k, or implementing test-time inference methods (https://arxiv.org/abs/2408.03314).
>
> > In general, well written paper. In some cases, the sentences are still a bit verbose and could be shortened. Exp.: In my opinion lines 42-60 are duplicative to the text in chapter 2, although I liked the bullet points. I acknowledge by being a physician that medical writing style is different than the style of the engineering sciences. Also paragraphs 5.2 and 5.3 are oddly sticking out. Please either drop them entirely, or elaborate more. If you decide to keep them, I don’t see the point in separating them. As such, 5.3 could be integrated into 5.2.
>
> We appreciate your positive feedback. We will proofread the paper and trim lengthy sentences. We can understand how section 5.3 “sticks out,” as it’s about tool learning whereas the other two sections are about datasets, specifically in medicine and mathematics. To fix this, we will adjust 5.3 to mention datasets similar to MedCalc-Bench. These datasets have been studied by researchers who investigated tool-learning solutions for LLMs to solve computation problems in the clinical setting.
>
> > It is a good paper, focusing on a very interesting and most likely widely used application of LLMs in the future. Good figures and tables in the paper enhance the readability. Room for improvement: Table 1 – please add legend for L., I suppose it should read length? Table 2 – is very busy with two decimals which suggest a non-existing level of accuracy, consider rounding to 1 decimal or even integers. Table 4 – even busier, consider rounding to 1 decimal. Please add explanation to error types as a legend for quick cross referencing.
>
> Thank you for your comments. Yes, you are correct, “L” stands for “length” in Table 1 and we will modify that column name. We will also round all reported values in Table 2 and Table 4 to one decimal point to make the tables more readable. Additionally, in the caption for Table 4, we will specify what each error type indicates by their exact name (Type A - Knowledge Errors, Type B - Extraction Errors, Type C - Computation Errors).
>
> We hope our response addresses your concerns and are pleased to answer any follow-up questions you may have.

---

> > ### Comment · Reviewer_aJHL · 2024-08-14
> > **Thank you for your revisions**
> >
> > Thank you for your response. All my concerns have been addressed. Hope you keep on adding data to your dataset.

---

> > > ### Author Response · Authors · 2024-08-14
> > >
> > > Thank you very much. We will maintain and improve our dataset on a regular basis.

---

### Official Review · Reviewer_An5a · 2024-07-15
**A new benchmark for medical calculation task.**

**Rating:** 7
**Confidence:** 3
**Correctness:** The dataset is constructed in a sound…
**Clarity:** The paper is general in good writing.

**Review:**

The paper is well organized and easy to read. Experimental results are adequate. Data and source code are available from GitHub.

**Strengths:**

1. A new task for medical calculations which is important for real-world setting.
2. Experiments are implemented in several sots LLMs including both open-source and close-source.

**Additional Feedback:**

Please refer to the above comments.

**Documentation:**

There is a GitHub link for the submitted dataset.

**Ethics:**

No ethical concern is found.

**Limitations:**

Refer to Opportunities For Improvement

**Opportunities For Improvement:**

1. It is not clear that how the evaluation of error type is realized by GPT4. And how human evaluation for such automation is implemented?
2. Error type sample shown in Figure 3. Are the color marked incorrect rationale and comments annotated by human or GPT4? If it is by GPT4, how is it realized?
3. I am not sure the std shown in Table2. Did you inference the model for several times?

**Relation To Prior Work:**

Yes the related works are discussed.

**Summary And Contributions:**

This paper proposed a new dataset for medical calculation task. The experimental results show that even GPT4 fail shot in correctly understand the patient note. The analysis for error type is also interesting.

---

> ### Author Rebuttal · Authors · 2024-08-14
>
> We thank the reviewer for their feedback and appreciate that they found the paper easy to read. To address your concerns:
>
> > 1. It is not clear how the evaluation of error type is realized by GPT4. And how human evaluation for such automation is implemented?
>
> The evaluation of error type by GPT-4 is done in two steps. Firstly, for the questions that the LLM incorrectly answers, we prompt GPT-4 to explain the mistakes in the LLM-generated explanation and then identify the error type (Type A - Knowledge Errors, Type B - Extraction Errors, Type C - Computation Errors, Type D - Other). This is done by giving the patient note, the calculation question, the LLM-generated explanation, the ground-truth explanation, and the error type definitions. We then randomly sampled 200 examples and manually checked whether GPT-4 had correctly identified the error type with the correct explanation for why the LLM-response is incorrect. We found an 89% agreement on this subset, which suggests that the GPT-4 error classifier is of high accuracy. As such, we used GPT-4 to classify the errors for all MedCalc-Bench questions  that an LLM gets incorrect and reported the numbers in Table 4.
>
> > 2. Error type sample shown in Figure 3. Are the color marked incorrect rationale and comments annotated by human or GPT4? If it is by GPT4, how is it realized?
>
> We assume that this comment relates to Table 3 as there is only one figure in the first 9 pages of our paper. The color-marked comments are annotated by humans. The purpose of this table is to provide some concrete examples of the three main error types by highlighting where the GPT-4-generated explanation is incorrect for a given MedCalc-Bench task.
>
> > 3. I am not sure the std shown in Table2. Did you inference the model for several times?
>
> We ran the inference for each model only once. The standard deviation reported in Table 2 is calculated as the proportion standard deviation sqrt(accuracy * (1- accuracy)/(total number of instances)). To help clarify this, we will add a caption to Table 2 on how the standard deviation is computed.
>
> We hope our response addresses your concerns and are pleased to answer any follow-up questions you may have.

---

> > ### Comment · Reviewer_An5a · 2024-08-15
> > **Thank you for your response**
> >
> > Most of my comment have been solved, thus improve my rate to 7.

---

> > > ### Author Response · Authors · 2024-08-15
> > >
> > > Thank you very much for raising the score. We appreciate your time and effort in reviewing our paper.

---

### Official Review · Reviewer_oeyh · 2024-07-20
**A Useful Benchmark for Evaluating Large Language Models in Biomedical Calculations**

**Rating:** 8
**Confidence:** 4
**Clarity:** This paper is well-written.

**Review:**

MedCalc-Bench addresses an important gap in evaluating the quantitative reasoning capabilities of LLMs in medicine. This paper makes a significant contribution by introducing a novel benchmark for medical calculations and providing valuable insights into the current capabilities and limitations of LLMs in this domain. The work highlights important areas for improvement in LLMs' quantitative reasoning abilities in medical contexts. Overall, it is a high-quality work with a clear presentation, well-organized and well-written, and significantly contributes to biomedical calculation by filling the gap of lacking such datasets and benchmarks.

**Strengths:**

1. Manually curated and verified dataset with explanations. They curated 55 common medical calculators from MDCalc, covering both rule-based and equation-based calculations.
2. Compiled patient notes from public sources and identified relevant notes for each calculator.
3. Generated step-by-step explanations and ground truth answers for each instance.
4. Comprehensive evaluation of multiple models and prompting strategies. They evaluated 8 different LLMs using zero-shot, zero-shot chain-of-thought, and one-shot chain-of-thought prompting.
5. Detailed error analysis providing insights for future improvements.

**Additional Feedback:**

See above.

**Correctness:**

The claims, datasets, and benchmarks, including evaluation methods and experiment design, were appropriate and performed correctly.

**Documentation:**

This dataset is well documented. The supplementary materials are very helpful.

**Ethics:**

No ethics review is needed.

**Limitations:**

See above.

**Opportunities For Improvement:**

1. Relatively small dataset size (1047 instances). I understand this is a pioneering work filling the gap, so I encourage the authors to keep maintaining and improving the datasets with more instances and calculators.
2. Consider evaluating few-shot learning beyond one-shot examples in future work.
3. For reproducibility, I would suggest the authors conduct experiments with the GPT APIs and set the temperature to 0.0.

**Relation To Prior Work:**

Related works are well discussed in general. Additional related works may include Almanac [1], OpenMedCalc [2], EHRAgent [3], etc.

[1] Zakka, Cyril, et al. "Almanac—retrieval-augmented language models for clinical medicine." NEJM AI 1.2 (2024): AIoa2300068.
[2] Goodell, Alex J., et al. "Augmentation of ChatGPT with Clinician-Informed Tools Improves Performance on Medical Calculation Tasks." medRxiv (2023): 2023-12.
[3] Shi, Wenqi, et al. "Ehragent: Code empowers large language models for complex tabular reasoning on electronic health records." arXiv preprint arXiv:2401.07128 (2024).

**Summary And Contributions:**

This paper Introduces MEDCALC-BENCH, a novel dataset for evaluating the medical calculation capabilities of large language models (LLMs). It contains over 1000 manually reviewed instances covering 55 different medical calculation tasks and provides comprehensive evaluations of various open and closed-source LLMs on this benchmark. In addition, it also analyzes error types to reveal knowledge gaps and deficiencies in LLMs for medical calculations.

---

> ### Author Rebuttal · Authors · 2024-08-14
>
> We thank the reviewer for their constructive feedback and positive comments on the presentation and novelty of our paper. To address your concerns:
>
> > Relatively small dataset size (1047 instances). I understand this is a pioneering work filling the gap, so I encourage the authors to keep maintaining and improving the datasets with more instances and calculators.
>
> We agree with you and have acknowledged that our test dataset is relatively small in the Limitations (Line 223-225), although this is comparable to commonly used biomedical benchmarks such as MedQA (1273 test instances) and PubMedQA (500 test instances). In addition to the high cost of domain-specific annotations, many calculators do not have enough eligible patients that are publicly available for instance curation. For example, while some calculators such as Adjusted Body Weight have several hundred patient notes available, this is not the case for less common calculators such as FENa and MeldNa Score. Hence,  we include up to 20 instances for a given calculator to create a balanced dataset that can fairly evaluate the clinical computation abilities of LLMs.
>
> We will keep maintaining the dataset. We also agree that adding more calculators will make it more comprehensive. Currently, MedCalc-Bench contains 55 calculators that are listed as “Popular” calculators on MDCalc. These calculators already cover a diverse computational and reasoning skill set (listed in Table 1). While calculators outside of this category could be useful as well, we did not find a systematic and unbiased way to select which ones to include in our study. For future work, we will add more calculators based on the interest of clinicians and the frequency of usage in the practice.
>
> > Consider evaluating few-shot learning beyond one-shot examples in future work.
>
> Thank you for your comment. Having more examples may improve LLMs' ability on MedCalc-Bench. Currently, the main challenge is that some calculators do not have enough notes for this to be possible. Additionally, the context of these notes can be thousands of tokens long and may not fit into the context length of some LLMs. We have acknowledged this limitation in our paper (Line 229-230). We will leave it as future work after more instances are curated for each calculator and longer-context LLMs are adopted in our evaluations.
>
> > For reproducibility, I would suggest the authors conduct experiments with the GPT APIs and set the temperature to 0.0.
>
> We agree with the importance of reproducibility, and all of our experiments were conducted with a temperature of 0 using the Azure OpenAI API. We will add this important detail to our evaluation settings.
>
> > Related works are well discussed in general. Additional related works may include Almanac [1], OpenMedCalc [2], EHRAgent [3], etc.
>
> Thank you for mentioning these relevant papers. We have discussed OpenMedCalc (Line 255) and will include the discussions of the other two studies.
>
> We hope our response addresses your concerns and are pleased to answer any follow-up questions you may have.

---

> > ### Comment · Reviewer_oeyh · 2024-08-14
> > **Thank you for your response**
> >
> > Thank you for your response. All my concerns have been addressed and I will keep my very positive score. I appreciate the merit and contribution of this benchmark, as well as potential impact to the larger community regarding medical AI and foundation models.

---

> > > ### Author Response · Authors · 2024-08-14
> > > **Thank you**
> > >
> > > We sincerely appreciate your time and effort in reviewing our paper.

---

### Decision · Program_Chairs · 2024-09-26

**Decision:**

Accept (Oral)

**Comment:**

This paper introduces MEDCALC-BENCH, a novel dataset designed to evaluate the medical calculation capabilities of large language models (LLMs). The dataset consists of over 1,000 manually reviewed instances spanning 55 distinct medical calculation tasks, along with comprehensive evaluations of various open and closed-source LLMs on this benchmark. Additionally, the paper provides an analysis of error types, highlighting knowledge gaps and deficiencies in LLMs when performing medical calculations. All reviewers agree that this paper makes a significant contribution to the biomedical field. However, it would be beneficial to continuously update the dataset, provide more detailed evaluations, and expand the current limitations. Overall, this is a high-quality paper, and I recommend its acceptance.